# Deceptive Humor: A Synthetic Multilingual Benchmark Dataset for Bridging Fabricated Claims with Humorous Content

## Abstract

In the evolving landscape of online discourse, misinformation increasingly adopts humorous tones to evade detection and gain traction. This work introduces Deceptive Humor as a new research direction, emphasizing how false narratives, when coated in humor, become more difficult to detect and more likely to spread. To support research in this space, we present the Deceptive Humor Dataset (DHD), a multilingual collection of humor-infused comments derived from fabricated claims using the ChatGPT-4o model. Each entry is annotated with a Satire Level (from 1 for subtle satire to 3 for overt satire) and categorized into five humor types: Dark Humor, Irony, Social Commentary, Wordplay, and Absurdity. The dataset spans English, Telugu, Hindi, Kannada, Tamil, and their code-mixed forms, making it a valuable resource for multilingual analysis. Building on this foundation, we propose DH-MTL (Deceptive Humor Multi-Task Learning), a lightweight neural framework that jointly models satire intensity and humor type through a two-stage training pipeline that first adapts the encoder to deceptive humor patterns and then refines task-specific reasoning. Together, DHD and DH-MTL establish both a benchmark resource and a methodological baseline for studying how false narratives are framed, normalized, or obscured through humor.

## 1 Introduction

**Caution:** The Paper Contains LLM-generated fabricated humor; reader discretion is advised.

In today's online world, humor is increasingly used as a wrapper around false claims, making misinformation appear lighthearted and harmless. We refer to this phenomenon as deceptive humor. At first glance, such comments seem harmless and entertaining, often making people laugh. However, beneath the playful tone, they can subtly embed and reinforce misinformation. Importantly, not all jokes that mention false claims fall into this category; satire, for instance, may openly mock or criticize misinformation. Deceptive humor works differently: it repeats or normalizes a false claim through humor, making the misinformation harder to notice and easier to accept. Because humor lowers people's guard, repeated exposure can shape beliefs without triggering careful scrutiny. Unlike traditional humor, which aims to entertain, deceptive humor masks fabricated narratives, weakening detection and accelerating their spread. While Appendix A highlights that current models struggle to detect deceptive humor, we argue that the challenge lies not merely in the humorous coating, but in the way deception is framed. To capture these nuances, the DHD annotates each comment along two complementary dimensions, Satire Level and Humor Attribute, reflecting the subtlety and stylistic strategies used to convey the false claim, as illustrated in Figure 1a.

Understanding this blend of humor and deception is essential for recognizing how fabricated claims can appear harmless when expressed jokingly[1]. To illustrate, consider the widely debunked claim: **"Ch\*na is spreading COVID as a bioweapon"** (see Appendix F). On the surface, humorous comments about this claim may seem playful and unrelated to the false narrative. Upon closer examination, a pattern emerges: some jokes simply exaggerate everyday experiences, such as *"Ch\*na products usually don't last long"* or *"My phone broke in a week, must be made in Ch\*na."* Others subtly link this everyday humor back to the false claim, for example: *"Ch\*na products usually don't*

---

[1]Anonymous links: Project Website, GitHub, and Dataset (The dataset will be released after acceptance.)

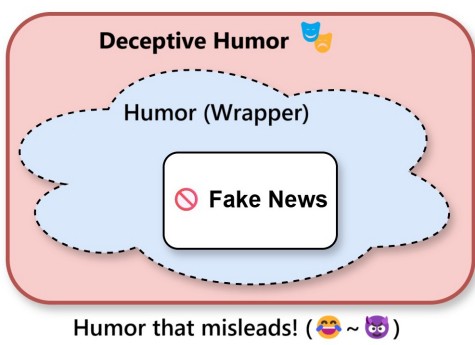 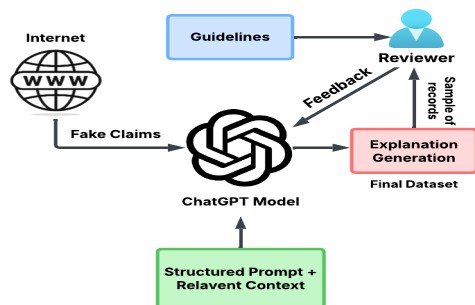

(a) Unmasking Deceptive Humor: Fake claims are embedded in humor, making them more engaging and harder to detect.

(b) Human-in-the-loop workflow for generating the Deceptive Humor Dataset through iterative feedback. (see Appendix I)

Figure 1: Illustration of the deceptive humor concept and DHD generation pipeline.

*last long, except this COVID thing"* or *"Guess Ch\*na finally made something that went global and stayed."* Rather than challenging the false claim, these comments keep the idea alive through humor. This contrast highlights how deceptive humor embeds fabricated narratives within ordinary jokes, making them appear casual while still echoing misinformation.

The literature provides substantial evidence that humor can actively contribute to the spread and believability of false claims, influencing how audiences perceive and propagate them (see Appendix G). Real-world observations further indicate that during crises, misinformation often appears in varied forms, with deceptive humor emerging as a particularly pervasive and influential channel (see Appendix H). These findings highlight the need to study how humor can subtly reinforce false claims.

While prior work has examined related phenomena such as Faux Hate (Biradar et al., 2024b;a), deceptive humor presents a distinct challenge. In our setting, deceptive humor refers to comments that embed a false claim within a humorous expression, not to mock the claim, but to restate or reinforce it in a playful tone. Because the humor masks the underlying narrative, such comments can appear harmless while still keeping the fabricated claim alive. Our dataset specifically targets this type of content: comments that project the false claim through humor, making it easier for users to consume without noticing the misinformation. To enable a systematic study of this phenomenon, we introduce the Deceptive Humor Dataset (DHD), a structured resource for analyzing how deceptive humor embeds false claims within playful expressions.

**Key Contributions:** In this work, we introduce the *Deceptive Humor Dataset (DHD)*, a novel multilingual resource that focuses specifically on humorous comments that embed a false claim in a playful tone, not to mock the claim, but to restate or normalize it. To the best of our knowledge, this is the first dataset designed to systematically study this form of humor, where misinformation is masked within a lighthearted expression. DHD provides detailed annotations for satire intensity and humor attributes. We further benchmark DHD using a diverse suite of pre-trained language models (PLMs), offering strong baselines for future work on fact-aware humor understanding. Additionally, we propose *DH-MTL*, a lightweight multi-task learning framework that jointly models satire intensity and humor type using ordinal regression and contrastive objectives with adaptive weighting, enabling the model to handle the ambiguity inherent in humorous expressions. Together, these contributions establish both the first dedicated resource and effective modeling tools for advancing research on deceptive humor detection.

## 2 LITERATURE REVIEW

Existing research often treats humor and misinformation as separate domains, leaving their intersection largely unexplored.

Table 1: Comparison of representative benchmark datasets in the domain of online discourse and social media safety across six key dimensions. The interdisciplinary attribute refers to the combination of two distinct domains within a single dataset, enabling the study of their interaction rather than analyzing them independently. ✓ indicates presence, while ✗ indicates absence.

| Dataset | Misinformation | Humor | Implicit/Explicit Hate | Fine-grained Cls | Multilingual | Interdisciplinary |
|---|---|---|---|---|---|---|
| LIAR (Wang et al., 2017) | ✓ | ✗ | ✗ | ✓ | ✗ | ✗ |
| FEVER (Thorne et al., 2018) | ✓ | ✗ | ✗ | ✗ | ✗ | ✗ |
| FakeNewsNet (Shu et al., 2020) | ✓ | ✗ | ✗ | ✗ | ✗ | ✗ |
| Hostile (Bhardwaj et al., 2020) | ✓ | ✗ | ✓ | ✓ | ✗ | ✗ |
| HaHackathon (Meaney et al., 2021) | ✗ | ✓ | ✗ | ✓ | ✗ | ✗ |
| Humicroedit (Hossain et al., 2019) | ✗ | ✓ | ✗ | ✓ | ✗ | ✗ |
| Memotion (Ramamoorthy et al., 2022) | ✗ | ✓ | ✗ | ✓ | ✗ | ✗ |
| Dark Humor (Kasu et al., 2025) | ✗ | ✓ | ✓ | ✓ | ✗ | ✗ |
| Implicit Hate (ElSherief et al., 2021) | ✗ | ✗ | ✓ | ✓ | ✗ | ✗ |
| Faux Hate (Biradar et al., 2024b) | ✓ | ✗ | ✓ | ✓ | ✗ | ✓ |
| **Deceptive Humor (Ours)** | ✓ | ✓ | ✓ | ✓ | ✓ | ✓ |

**Humor:** Humor has a dual role in social and psychological contexts. The Interpersonal Humor Deception Model (IHDM) suggests that humor can either reduce self-centered deception and foster trust or, when misused, raise suspicion and undermine credibility (Gaspar et al., 2023). Humor is widely used in communication, including advertising, where it can mask deceptive practices; studies report that a significant portion of humorous advertisements contain misleading elements that obscure unethical messaging (Shabbir et al., 2007). Computational research has largely focused on sarcasm (Joshi et al., 2015), irony (Van Hee et al., 2018), and satire (Rubin et al., 2016), with datasets such as HaHackathon (Meaney et al., 2021), Dark Humor (Kasu et al., 2025), Humicroedit (Hossain et al., 2019), and Memotion (Ramamoorthy et al., 2022) capturing linguistic or stylistic humor features. However, these resources do not consider humor grounded in fabricated claims, limiting their utility for detecting deceptive humor.

**Misinformation:** Detection of misinformation has largely focused on textual veracity, social propagation, and user behavior (Thorne et al., 2018; Wang et al., 2017; Shu et al., 2020; Bhardwaj et al., 2020). While effective in identifying false or misleading content, these approaches do not account for how humor can distort facts, creating nuanced forms of deception. Existing frameworks and datasets mostly treat misinformation independently, without examining how humorous presentation interacts with credibility, context, or factual distortion.

Despite these advances, current benchmarks fail to jointly consider humor, misinformation, and harmful content, making it difficult to study how humor can serve as a medium for deception in online discourse. As shown in Table 1, existing datasets largely focus on single dimensions, whereas the proposed Deceptive Humor Dataset (DHD) integrates multiple dimensions, including satire, irony, misinformation, and implicit/explicit hate, enabling the first comprehensive analysis of deceptive humor in multilingual social media contexts.

## 3 DATASET DEVELOPMENT

In this section, we describe the process of selecting fabricated claims and generating the Deceptive Humor Dataset (DHD). Using ChatGPT-4o, we create humor-infused comments across multiple languages, ensuring diversity in satire and linguistic variations. We also highlight the role of synthetic data in advancing AI (see Appendix J), emphasizing its importance in training robust models and addressing data scarcity in multilingual settings.

### 3.1 SELECTION OF FAKE CLAIMS

The first step in data acquisition involves identifying a wide range of topics to ensure diversity in the collected data. The authors selected themes such as entertainment, politics, finance, sports, religion, and health. After selecting the topical domains, the next step is to identify fabricated narratives associated with each of them. These narratives are systematically scraped from reputable fact-checking platforms such as AltNews, Boom FactCheck, FactChecker, and FACTLY. This process ensures that the deceptive claims used as inputs for humor generation are both reliable and grounded in verified misinformation cases.

## 3.2 GENERATION OF DECEPTIVE HUMOR CORPUS

In this section, we describe the construction of the Deceptive Humor Dataset (DHD) using ChatGPT-4o (Hurst et al., 2024). Deceptive humor is inherently complex and difficult to collect reliably at scale, as such comments often blend subtly into normal discourse and require contextual knowledge to identify. To address this, we leverage the controllability and scalability of LLMs to generate high-quality, humor-infused comments grounded in fabricated claims. This approach ensures consistency across multiple languages while maintaining diversity in humor styles and linguistic variations.

Generating humor with embedded deception is challenging, requiring a balance between satire and subtle misinformation. We evaluated several state-of-the-art generative models, including Gemini (Team et al., 2023), LLaMA (Touvron et al., 2023), Claude, and ChatGPT. While Gemini and LLaMA perform reasonably for English, they often produce ungrammatical outputs in Indic languages; Claude frequently declines requests involving humor and deception. ChatGPT-4o consistently generates coherent, contextually appropriate, and humorous comments across languages. The generated corpus was further reviewed by language experts to ensure natural, engaging, and safe content (see subsection I.2 for structured prompting details).

To ensure high-quality and reliable data, we adopt a human-in-the-loop (HITL) workflow in which LLM-generated comments are iteratively refined using structured prompts, expert feedback, and contextual examples. Because deceptive humor is challenging and synthetic text often diverges from real-world patterns, we used a batchwise review process (100 samples per iteration) to inspect most outputs and correct common failure modes. The review process involved three postgraduate students, two PhD students with expertise in hate speech and computational humor, and a supervising professor who guided the overall procedure. This structured prompting and feedback loop, shown to significantly enhance quality in HITL literature, helped reduce typical synthetic-data errors and align the outputs more closely with real-world deceptive humor. These controls ensured consistency, quality, and safety across batches (for the detailed HITL workflow we followed, see Appendix I). The overall process is illustrated in Figure 1b.

## 3.3 DATASET DESCRIPTION

The proposed DHD consists of 9,000 synthetically generated humorous comments, carefully curated to ensure linguistic diversity and humor variation. The dataset is split into 7,200 comments for training, 900 for validation, and 900 for testing. Each comment is labelled with a satire level ranging from 1 to 3, where 1 represents subtle satire and 3 denotes highly exaggerated satire. Additionally, every comment is assigned a humor attribute from one of the five predefined categories: Irony, Absurdity, Social Commentary, Dark Humor, and Wordplay.

A key aspect of the DHD is its linguistic diversity. Along with English, it includes comments in four major Indic languages: Telugu, Hindi, Kannada, and Tamil, along with their code-mixed versions. This ensures a rich and varied dataset that captures the nuances of humor across multiple languages and cultural contexts. The structured labeling enables a comprehensive analysis of humor in NLP systems, fostering advancements in computational humor understanding, particularly in multilingual and code-mixed settings. A detailed description of the dataset is presented in Table 2.

## 3.4 TASKS SUPPORTED BY THE DECEPTIVE HUMOR DATASET

The Deceptive Humor Dataset (DHD) models how misinformation is framed and delivered through humor. Since every comment in DHD is anchored in a fabricated claim, the key variation lies not in the presence of deception but in how humor shapes its presentation, believability, and detectability. To capture this, we annotate two complementary dimensions, Satire Level and Humor Attribute, reflecting the degree of subtlety or exaggeration and the stylistic strategies used to express the false claim. These annotations focus on how misinformation is humorously framed, rather than humor in isolation, enabling detailed analysis of the mechanisms by which false claims are normalized or obscured. Although the tasks do not explicitly identify misinformation, they provide structured supervision for modeling the pathways through which humor makes fabricated content persuasive.

Table 2: Distribution of satire levels and humor attributes across languages. (* indicates Indic languages along with their code-mixed variants.) Humor attribute abbreviations: Abs. = Absurdity, Dark = Dark Humor, Irony = Irony, Soc. Comm. = Social Commentary, Word. = Wordplay. (For summary see Table 10)

| Language | Total | Satire 1 | Satire 2 | Satire 3 | Abs | Dark | Irony | Soc. Comm. | Word. |
|---|---|---|---|---|---|---|---|---|---|
| **Train Data** | | | | | | | | | |
| English | 881 | 188 | 322 | 371 | 259 | 149 | 292 | 101 | 80 |
| Telugu* | 1663 | 494 | 698 | 471 | 372 | 275 | 552 | 274 | 190 |
| Hindi* | 1624 | 480 | 735 | 409 | 284 | 204 | 510 | 316 | 310 |
| Kannada* | 1536 | 461 | 674 | 401 | 397 | 262 | 384 | 285 | 208 |
| Tamil* | 1496 | 457 | 709 | 330 | 349 | 198 | 462 | 239 | 248 |
| **Validation Data** | | | | | | | | | |
| English | 114 | 25 | 37 | 52 | 22 | 25 | 37 | 14 | 16 |
| Telugu* | 221 | 68 | 99 | 54 | 47 | 33 | 67 | 35 | 39 |
| Hindi* | 195 | 59 | 86 | 50 | 35 | 26 | 52 | 34 | 48 |
| Kannada* | 198 | 67 | 86 | 45 | 47 | 30 | 62 | 39 | 20 |
| Tamil* | 172 | 57 | 74 | 41 | 29 | 22 | 64 | 33 | 24 |
| **Test Data (Human Annotated)** | | | | | | | | | |
| English | 104 | 38 | 32 | 34 | 33 | 15 | 30 | 7 | 19 |
| Telugu* | 204 | 112 | 64 | 28 | 73 | 21 | 57 | 20 | 33 |
| Hindi* | 207 | 37 | 26 | 143 | 27 | 36 | 76 | 14 | 22 |
| Kannada* | 189 | 67 | 40 | 82 | 62 | 40 | 39 | 20 | 28 |
| Tamil* | 196 | 113 | 64 | 19 | 51 | 21 | 57 | 42 | 25 |

### 3.4.1 TASK 1: SATIRE INTENSITY CLASSIFICATION

This task involves classifying the intensity of satire in a given comment, which is a crucial proxy for the subtlety of the underlying deception. The goal is to predict one of three ordinal levels, reflecting the degree to which humor is used to mask a false claim.

- **Low Satire:** The humor is subtle and lightly satirical, often resembling real-world statements with a mild twist.

- **Moderate Satire:** The humor is more evident, incorporating exaggeration and sarcasm while maintaining a balance between reality and absurdity.

- **High Satire:** The humor is strongly exaggerated and overtly satirical, often making use of extreme irony or absurd distortions of reality.

### 3.4.2 TASK 2: HUMOR ATTRIBUTE CLASSIFICATION

This task requires models to categorize the specific type of humor in a comment. Unlike Satire Level, which measures intensity, Humor Attribute focuses on stylistic and rhetorical devices. Although some categories may overlap, maintaining fine-grained distinctions is essential to support nuanced analysis and robust model evaluation.

- **Irony**: A form of humor where the intended meaning contrasts sharply with the literal meaning, often exposing contradictions or unexpected outcomes.

- **Absurdity**: Humor that thrives on exaggeration, illogical scenarios, or unrealistic premises to create an amusing effect.

- **Social Commentary**: Humor that critiques, mocks, or highlights societal or cultural issues, often with a satirical or thought-provoking angle.

- **Dark Humor**: Humor that deals with morbid, taboo, or controversial topics in a way that might be unsettling but still amusing.

- **Wordplay**: Humor that relies on clever linguistic constructs, including puns, double meanings, and phonetic playfulness.

## 4 METHODOLOGY

We introduce **DH-MTL** (*Deceptive Humor Multi-Task Learning*), a lightweight neural framework designed specifically for the challenges of deceptive humor detection. Unlike traditional humor datasets, DHD contains two complementary but non-trivial annotation dimensions: (i) *Satire Level*, which is inherently ordinal, and (ii) *Humor Attribute*, which is categorical but often ambiguous due to overlapping humor styles. In addition, deceptive humor deliberately blends exaggeration and subtlety, requiring models to be both robust and uncertainty-aware. To meet these challenges, DH-MTL employs a two-stage training framework for better domain adaptability.

**Shared Transformer Encoder**

We employ a pretrained Transformer $f_\theta$ (BERT-large) to encode each comment. Given input text $x$, the encoder produces a pooled representation

$$\mathbf{h} = f_\theta(x),$$

which captures contextual markers of exaggeration, irony, incongruity, and pragmatic misdirection intrinsic to deceptive humor. This representation is shared across both tasks.

**Satire Level as an Ordinal Prediction Task**

Satire Level consists of three ordered intensities $\{0, 1, 2\}$. Because the degree of humorous distortion is inherently monotonic (i.e., high satire always implies at least moderate satire), we adopt an ordinal regression formulation. Instead of learning three independent logits, we model $K-1 = 2$ cumulative comparisons:

$$P(y > k) = \sigma(z_k), \quad k \in \{0, 1\}.$$

Each $z_k$ is computed by a lightweight satire head $g_{\text{sat}}(\mathbf{h})$. Let $t_{ik} = \mathbb{1}\{y_i > k\}$ denote threshold labels. The satire objective is

$$\mathcal{L}_{\text{sat}} = \sum_{i=1}^{N} \sum_{k=0}^{K-2} \text{BCE}(\sigma(z_{ik}), t_{ik}),$$

which enforces monotonicity and preserves ordinal structure. This is crucial: intensity progression from subtle satire to overt absurdity is continuous, and ordinal modeling captures this continuum more faithfully than flat classification.

**Humor Attribute as Multi-Class Classification**

While satire intensity is ordered, Humor Attribute expresses stylistic differences with no inherent ranking (e.g., irony, absurdity, wordplay). We therefore use a standard softmax classifier $g_{\text{hum}}(\mathbf{h})$:

$$p = \text{softmax}(g_{\text{hum}}(\mathbf{h})).$$

Because humorous styles are imbalanced in natural discourse, we employ a focal loss to prevent dominance by frequent humor types and promote better discrimination across stylistic subtleties:

$$\mathcal{L}_{\text{hum}} = -\sum_{i=1}^{N} \sum_{c=1}^{C} (1 - p_{ic})^\gamma \, y_{ic} \log(p_{ic}).$$

This enhances learning from difficult or infrequent humor mechanisms.

### 4.1 JOINT MULTI-TASK OPTIMIZATION WITH TWO-STAGE TRAINING

Deceptive humor presents a dual challenge: it combines stylistic variations with varying intensity levels, and its linguistic patterns differ substantially from general-domain text. To address this, DH-MTL jointly optimizes both tasks using a weighted combination of the Satire Level and Humor Attribute losses:

$$\mathcal{L}_{\text{total}} = w_{\text{sat}} \, \mathcal{L}_{\text{sat}} + w_{\text{hum}} \, \mathcal{L}_{\text{hum}}.$$

The weighting coefficients are dynamically updated during training to reflect evolving task difficulty, preventing one task from dominating and allowing the model to allocate capacity adaptively between intensity and stylistic cues. This joint optimization is embedded within a two-stage training pipeline: Stage 1 performs domain adaptation to align the pretrained encoder with the unique distribution of deceptive humor, while Stage 2 fine-tunes task-specific heads with gradual unfreezing, enabling the model to refine both satire intensity and humor style predictions. Together, these design choices ensure balanced multi-task learning while capturing the nuanced characteristics of deceptive humor.

**Stage 1: Domain Adaptation.** The primary motivation for domain adaptation stems from the nature of deceptive humor: it is a relatively new problem with no readily available pretrained models specifically tailored for its linguistic patterns. Existing language models are trained on general domain text and may not capture the nuanced exaggeration, irony, or culturally grounded references inherent to deceptive humor. In this stage, all encoder layers and both task-specific heads are jointly optimized. The goal is not yet high task precision, but rather to align the pretrained Transformer with the unique distribution of deceptive humor. Training both tasks simultaneously encourages the shared encoder to internalize the interaction between *how strongly* a claim is distorted and *which stylistic mechanism* delivers that distortion, establishing a domain-aligned representation space that captures deceptive humor holistically.

**Stage 2: Task-Specific Fine-Tuning.** Once domain adaptation stabilizes, we refine task specialization through gradual unfreezing. Beginning with only the top layers of the encoder trainable, and progressively unfreezing deeper layers over successive phases, the model transitions from broad domain alignment to precise task reasoning. This curriculum achieves two outcomes: (1) it preserves foundational linguistic knowledge while allowing deeper layers to adapt to deception-specific patterns, and (2) it encourages the satire and humor heads to carve out separable yet related subspaces within the shared encoder. This stage greatly improves the model's ability to distinguish fine-grained humorous styles and subtle differences in satire intensity.

DH-MTL explicitly models the unique properties of deceptive humor. Dynamic loss weighting balances satire intensity and humor style, ordinal regression preserves intensity ordering, and the two-stage training pipeline combines domain alignment with task-specific fine-tuning. Together, these components produce structured representations that enable better generalization to unseen examples compared to single-task or static multi-task baselines.

# 5 RESULTS AND DISCUSSION

We evaluated a diverse set of model architectures: Encoder-Only, Encoder-Decoder, and LLMs (Brown et al., 2020) across Zero-Shot, Few-Shot, and QLoRA-based (Dettmers et al., 2024) fine-tuning settings, for both Satire Level classification and Humor Attribute prediction. Among the LLMs, LLaMA-3.2-3B-Instruct outperformed most of the baselines in Satire Level prediction, demonstrating a clear dominance in capturing satire nuances. For Humor Attribute prediction, the LLaMA-3.1-8B-Instruct model achieved comparatively better performance than the other baselines. While LLMs struggled with Zero-Shot and Few-Shot settings, often failing to correctly identify labels in deceptive humor comments, QLoRA fine-tuning significantly improved their understanding, enabling them to compete closely with, or sometimes even outperform, Encoder-Only and Encoder-Decoder based models (Table 3).

These findings underscore the challenges LLMs face with nuanced tasks like deceptive humor classification, which requires deep contextual, cultural, and factual understanding. The subtlety and ambiguity of deceptive humor often lead to misclassifications or omission of certain classes. Prior work, such as the Memotion analysis task (Sharma et al., 2020), has shown that fine-grained humor detection becomes increasingly difficult, especially with limited modalities. Consistently, our experiments reveal that using only the text modality significantly reduces performance, highlighting the open nature of this research problem. A detailed error analysis and discussion of challenging samples are provided in Appendix L, offering insights into current limitations and directions for future work.

Existing models, while effective on humor detection or misinformation classification individually, struggle with deceptive humor, which requires both fact verification and intent recognition. Prior

datasets such as SemEval-2017 Task 6 (Potash et al., 2017), Humicroedit (Hossain et al., 2019), FEVER (Thorne et al., 2018), LIAR (Wang et al., 2017), and Hostile (Bhardwaj et al., 2020) address either humor or misinformation but not their intersection. Even synthetic humor datasets, like Unfun (Horvitz et al., 2024), focus on humor manipulation rather than humor entwined with deception.[2]

Our proposed DH-MTL framework addresses this gap by jointly modeling Satire Level (ordinal) and Humor Attribute (categorical) in a multi-task setup. As shown in Table 3, DH-MTL excels in modeling the ordinal structure of Satire Level, achieving the highest Pearson correlation (33.57) on the DHD dataset. The model balances learning across all classes and maintains clear distinctions between overlapping Humor Attribute labels, capturing fine-grained stylistic differences. Together, dynamic loss weighting and joint multi-task learning enable DH-MTL to provide more reliable predictions for both satire intensity and humor style compared to traditional baselines.

Table 3: Baseline metrics of models across satire levels and humor attributes. Metrics include Accuracy (Acc), Macro F1 (MacF1), Weighted F1 (WgtF1), and Pearson Correlation (Pear). Top results are in **bold**,while the second-best performance is marked with a [†].

| Model | Satire Level | | | | Humor Attribute | | |
|---|---|---|---|---|---|---|---|
| | Acc | MacF1 | WgtF1 | Pear | Acc | MacF1 | WgtF1 |
| **Encoder-Only** | | | | | | | |
| BERT (Devlin et al., 2019) | 45.33 | 44.06 | 44.47 | 28.10 | 35.44 | 30.09 | 32.33 |
| DistilBERT (Sanh et al., 2019) | 42.00 | 40.43 | 40.80 | 22.20 | 32.00 | 28.02 | 29.97 |
| mBERT | 45.56 | 44.97 | 45.20 | 27.64 | 33.78 | 31.37 | 32.82 |
| XLM-RoBERTa (Conneau et al., 2019) | 43.11 | 41.15 | 41.66 | 26.69 | 34.00 | 30.99 | 32.57 |
| DeBERTa (He et al., 2020) | 41.22 | 30.15 | 32.60 | 20.19 | 31.22 | 28.20 | 29.56 |
| **Encoder-Decoder** | | | | | | | |
| BART (Lewis et al., 2020) | 42.33 | 39.65 | 40.35 | 19.17 | 35.33 | 31.73 | 33.54 |
| T5 (Raffel et al., 2020) | 42.33 | 35.31 | 36.67 | 20.75 | 28.44 | 11.91 | 15.11 |
| **Decoder-Only (Zero-Shot)** | | | | | | | |
| Gemma-2-2b-it (Team et al., 2024) | 32.00 | 27.66 | 30.24 | 13.47 | 20.56 | 18.97 | 22.50 |
| Llama-3.2-3b-it (Grattafiori et al., 2024) | 38.44 | 33.07 | 35.22 | 19.23 | 24.89 | 20.44 | 25.10 |
| Mistral-7b-v0.3-it (Jiang et al., 2023) | 31.35 | 29.52 | 29.47 | 12.12 | 27.25 | 27.00 | 26.92 |
| Llama-3.1-8b-it (Grattafiori et al., 2024) | 34.00 | 29.83 | 29.69 | 13.37 | 25.72 | 23.89 | 23.97 |
| **Decoder-Only (Few-Shot)** | | | | | | | |
| Gemma-2-2b-it | 34.73 | 30.80 | 32.25 | 18.55 | 22.56 | 14.87 | 18.75 |
| Llama-3.2-3b-it | 38.78 | 35.27 | 36.12 | 18.79 | 27.55 | 25.55 | 28.20 |
| Mistral-7b-v0.3-it | 30.55 | 30.29 | 31.97 | 15.40 | 28.57 | 28.44 | 30.20 |
| Llama-3.1-8b-it | 37.55 | 33.91 | 34.20 | 17.51 | 30.25 | 28.91 | 29.57 |
| **Decoder-Only (QLoRA Fine-Tuned)** | | | | | | | |
| Gemma-2-2b-it | 39.22 | 36.49 | 40.27 | 28.72 | 27.89 | 30.31 | 31.22 |
| Llama-3.2-3b-it | **48.67** | **47.84** | 47.39[†] | 32.91[†] | 34.25 | 30.42 | 32.25 |
| Mistral-7b-v0.3-it | 40.67 | 33.89 | 35.14 | 13.18 | 27.11 | 18.53 | 18.56 |
| Llama-3.1-8b-it | 48.33[†] | 47.82[†] | **48.03** | 31.67 | 36.22[†] | **35.57** | **36.01** |
| **Proposed Work** | | | | | | | |
| DH-MTL (Ours) | 46.22 | 42.89 | 43.53 | **33.57** | **36.56** | 31.82[†] | 34.00[†] |

## 5.1 Ablation and Parameter Sensitivity analysis

To assess the importance of dynamic loss weighting, we conducted an ablation study with three configurations: (1) fixing the Satire weight ($w_{sat} = 1$), (2) fixing the Humor weight ($w_{hum} = 1$), and (3) fixing both weights ($w_{sat} = w_{hum} = 1$). As shown in Appendix D and discussed in subsection D.1, these experiments demonstrate that adaptive weighting is essential for balancing the two tasks, ensuring robust Satire Level predictions and fine-grained Humor Attribute distinctions. This validates the critical role of dynamic loss adjustment in DH-MTL's multi-task learning.

---

[2]ColBERT Humor Detection Dataset: https://huggingface.co/datasets/CreativeLang/ColBERT_Humor_Detection

DH-MTL incorporates two trainable parameters in the multi-task loss: the dynamic weighting coefficients for Satire Level ($w_{sat}$) and Humor Attribute ($w_{hum}$). To evaluate their impact, we conducted a parameter sensitivity analysis by varying batch size and learning rate, observing that the model remains stable across reasonable settings. This demonstrates that the dynamic weighting effectively balances the two tasks without introducing instability. For a full analysis, see Appendix E.

## 5.2 HUMAN EVALUATION

To ensure data quality, the DHD test set (900 samples) was manually annotated by the authors, leveraging their familiarity with the fabricated claims and the nuanced ways in which humor embeds deception. Annotating deceptive humor is inherently challenging, as it requires recognizing subtle fabrications, implied meanings, and culturally grounded cues that non-expert annotators often miss. A mock annotation round was conducted to refine guidelines and resolve ambiguous cases, after which consistent labeling procedures were followed for Satire Level and Humor Attribute. This author-driven annotation ensured that both surface-level humor and deeper contextual implications were accurately captured, providing a reliable benchmark for evaluating model performance.

Inter-annotator agreement results (see Appendix K for extended human evaluation details, Table 8) indicate fair to moderate alignment for Satire Level and moderate to substantial alignment for Humor Attributes, with strongest agreement observed in English and slightly lower but stable agreement across Indic languages and their code-mixed variants. Complementary human quality assessments (see Appendix K, Table 9) show consistently high readability and claim-graspability across languages, with cultural nuance best preserved in English and moderately captured elsewhere. These results establish DHD as a dependable benchmark for multilingual deceptive-humor modeling. The dataset was curated through rigorous human-in-the-loop procedures that actively enforce ethical safeguards and explicitly account for known limitations, including synthetic content and cultural coverage (see Appendix B and Appendix C).

## 6 CONCLUSION AND FUTURE WORK

In this study, we highlighted a real-world phenomenon at the intersection of misinformation and humor that has been largely overlooked in existing research, emphasizing how humor can serve as a major vehicle for spreading misinformation. Our approach underscores the need to better understand the interplay between humor and misinformation, an aspect often neglected in prior studies. To facilitate this exploration, we introduced the Deceptive Humor Dataset (DHD), the first multilingual and code-mixed benchmark for deceptive humor, along with strong baselines across pre-trained language models and large language models. Complementing the dataset, We propose DH-MTL, a lightweight two-stage multi-task framework that first adapts the encoder to deceptive humor patterns and then fine-tunes task-specific heads to jointly model Satire Level intensity and Humor Attributes. This design effectively captures subtle satire and fine-grained humor styles, providing robust, balanced predictions and demonstrating clear advantages over traditional single-task or static multi-task approaches. Together, DHD and DH-MTL establish both a benchmark resource and a methodological foundation for future research.

Looking forward, we plan to host an open leaderboard to encourage systematic benchmarking and foster community participation. In addition, we are developing DHD-HARD, a hidden and more challenging test split. Unlike the current dataset, which is fully automated with a human-in-the-loop process, DHD-HARD will be hybrid in nature, combining deceptive humor comments generated by LLMs with human-authored comments. Building on the controlled foundation of DHD, the proposed DHD-HARD will extend evaluation to noisy, in-the-wild environments, enabling a rigorous study of the sim-to-real gap. Moreover, our empirical findings reinforce the difficulty of this task, as even state-of-the-art LLMs often failed to detect deceptive humor in zero-shot and few-shot settings, underscoring the necessity of specialized benchmarks like DHD. By adopting a human-in-the-loop generation pipeline, we ensured high-quality and culturally grounded data across languages, further enhancing the reliability of our resource. Ultimately, this work highlights not only the technical but also the societal importance of studying deceptive humor, given its ability to subtly influence public beliefs while remaining disguised as entertainment. While the current dataset primarily focuses on fabricated claims grounded in Indian contexts, future work aims to expand DHD to capture deceptive humor at a global scale, incorporating broader cultural nuances and region-specific subtleties.

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

# A  LLM-BASED EVALUATION OF DECEPTIVE HUMOR DETECTION

**Motivation**

Deceptive Humor (DH) is a form of content where humor acts as a vehicle for misinformation, allowing misleading or false information to propagate more effectively in online and social media platforms. The subtlety of DH makes detection challenging, even for instruction-tuned LLMs. Unlike standard fake news or hate speech, the humorous coating masks the underlying deception, increasing virality and impact.

To empirically study this phenomenon, we evaluate both widely-used Small Language Models (SLMs) and state-of-the-art Large Language Models (LLMs) on our curated Deceptive Humor Dataset (DHD), which consists of 9,000 multilingual and code-mixed comments, all of which are inherently deceptive. Instead of directly performing full classification of the type or level of deceptive humor, we first probe whether the models can recognize at a basic level whether a given comment appears deceptive or not. This step assesses the models' initial sensitivity to deceptive cues. Our experiment findings show that even this basic detection task remains difficult for both SLMs and LLMs.

Our experimental findings reveal several important insights regarding the detection of Deceptive Humor:

- **Small Language Models (SLMs):** In our experiments, only LLaMA-3.2-3B Instruct achieved moderate performance, with ∼33% overall accuracy. Its performance was notably lower on code-mixed languages, highlighting the challenge of linguistic complexity. Qwen-2.5-3B Instruct failed to identify any deceptive comments, achieving 0% accuracy overall. These results indicate that smaller models struggle to detect deceptive humor effectively.

- **Large Language Models (LLMs):** LLaMA-3 8B Instruct achieved the highest overall accuracy (∼67%), although performance varied across languages and code-mixed contexts. Qwen-2.5 7B Instruct achieved 14% accuracy overall, showing that even large models face difficulty identifying deceptive humor, particularly in non-English and script-based languages.

- **Task Complexity:** Overall, our experiments demonstrate that detecting deceptive humor is inherently difficult. Model performance varies substantially across languages and code-mixed contexts, emphasizing that the combination of humor, deception, and linguistic diversity complicates detection.

Collectively, these results empirically support our claim that Deceptive Humor significantly complicates misinformation detection and facilitates wider propagation of misleading content (see Table 4 and Table 5 for detailed model-wise performance).

## A.1  EXPERIMENTAL PROMPT USED

---

**Prompt Used in our Experiments**

You are an expert in identifying fake news and deceptive content.
You will be given a comment. Your task is to determine if the comment is deceptive.

**Important instructions:**
- Respond strictly with either "Yes" if the comment is surely fake/deceptive, or "No" if it is not.
- Do not write or explain anything else.
- Do not include quotes or punctuation, just 'Yes' or 'No'.

**Comment:** {comment}

**Your Answer:**

---

Table 4: Performance of Small Language Models on the Deceptive Humor dataset (3B models).

| Language | LLaMA-3.2 3B Instruct | | | Qwen-2.5 3B Instruct | | |
|---|---|---|---|---|---|---|
| | Total | Identified | Accuracy | Total | Identified | Accuracy |
| Te | 1049 | 386 | 0.368 | 1049 | 0 | 0.00 |
| Te-En | 1039 | 266 | 0.256 | 1039 | 0 | 0.00 |
| Hi | 1002 | 340 | 0.339 | 1002 | 0 | 0.00 |
| Hi-En | 1024 | 306 | 0.299 | 1024 | 0 | 0.00 |
| Ka | 936 | 359 | 0.384 | 936 | 0 | 0.00 |
| Ka-En | 987 | 276 | 0.280 | 987 | 0 | 0.00 |
| En | 1099 | 446 | 0.406 | 1099 | 0 | 0.00 |
| Ta | 913 | 328 | 0.359 | 913 | 0 | 0.00 |
| Ta-En | 951 | 251 | 0.264 | 951 | 0 | 0.00 |
| **Overall** | 9000 | 2958 | **0.33** | 9000 | 0 | **0.00** |

Table 5: Performance of Large Language Models on the Deceptive Humor dataset (7B and 8B models).

| Language | LLaMA-3 8B Instruct | | | Qwen-2 7B Instruct | | |
|---|---|---|---|---|---|---|
| | Total | Identified | Accuracy | Total | Identified | Accuracy |
| Te | 1049 | 644 | 0.614 | 1049 | 63 | 0.060 |
| Te-En | 1039 | 756 | 0.728 | 1039 | 104 | 0.100 |
| Hi | 1002 | 685 | 0.684 | 1002 | 217 | 0.217 |
| Hi-En | 1024 | 791 | 0.772 | 1024 | 380 | 0.371 |
| Ka | 936 | 453 | 0.484 | 936 | 76 | 0.081 |
| Ka-En | 987 | 703 | 0.712 | 987 | 57 | 0.058 |
| En | 1099 | 872 | 0.793 | 1099 | 196 | 0.178 |
| Ta | 913 | 459 | 0.503 | 913 | 70 | 0.077 |
| Ta-En | 951 | 686 | 0.721 | 951 | 68 | 0.072 |
| **Overall** | 9000 | 6049 | **0.67** | 9000 | 1231 | **0.14** |

## A.2 POTENTIAL CHALLENGES FOR LLMS IN DETECTING DECEPTIVE HUMOR

Our zero- and few-shot results (Appendix A, Table 4, Table 5) show that instruction-tuned LLMs (e.g., Qwen-2.5 3B Instruct, **0% accuracy**) largely fail on this task. Several factors explain this gap:

- **Hidden Pragmatic Intent.** False claims are conveyed indirectly via irony or wordplay, so surface forms often resemble literal statements.
- **Lack of Context.** Many comments depend on conversational history, cultural cues, or multimodal context absent in isolated text snippets.
- **Cultural and Linguistic Gaps.** Code-mixing, idioms, and regional references are underrepresented in pretraining, leading to systematic misinterpretation.
- **Domain Shift.** Pretraining rarely includes humor deliberately embedding false claims, creating a sharp distributional mismatch.

Together, these factors show why even large instruction-tuned models cannot reliably detect deceptive humor without task-specific adaptation.

## B    ETHICAL CONSIDERATIONS

**Misuse Potential and Mitigation:** The Deceptive Humor Dataset (DHD) contains humorous expressions that embed, restate, or normalize fabricated claims, including politically sensitive narratives. Such content introduces dual-use risks, such as enabling systems that generate or amplify misleading or harmful messages under the guise of humor. To mitigate these risks, DHD is released only under controlled access with strictly prohibited redistribution. Access is limited to verified researchers who formally agree not to use the dataset or any derivative models to train generative systems, deploy misinformation-related applications, or create humor that repeats, amplifies, or legitimizes false claims. Compliance is enforced through verification of access requests and adherence to the signed data usage agreement.

**Privacy and Legal Compliance:** The humorous comments in DHD are synthetically generated using the ChatGPT-4o model. While the underlying fabricated claims were curated from public fact-checking archives (e.g., AltNews) under Fair Use principles, the humorous text itself is fully synthetic and not derived from real user posts.. As a result, the dataset contains no personal identifiable information (PII), no user metadata, and no material subject to third-party copyright or platform Terms of Service. Because the content is fully synthetic, there is no risk of re-identification. The dataset is released solely for non-commercial research purposes.

**Bias and Content Warning:** Because the dataset contains humorous content referencing fabricated or sensitive narratives, it may include text that could be perceived as offensive, biased, or harmful toward specific groups. We caution researchers to interpret these examples responsibly and explicitly advise against using this dataset to train or evaluate systems that may automate hate speech, harassment, or politically manipulative messaging.

**Responsible Annotation:** Annotation was mainly carried out by three postgraduate student assistants under the supervision of two PhD researchers and faculty members who provided informed consent prior to beginning the task. Annotators were compensated above the local minimum wage and were given clear task guidelines, including the right to opt out of any example containing sensitive or disturbing content. To protect annotators' well-being, we imposed a hard limit of 150 items per day and encouraged regular breaks throughout each session. Participation could be discontinued at any time without penalty. As an additional academic benefit, annotators were offered up to 150 hours of GPU compute for their own independent research projects; this benefit was optional and not tied to annotation performance. The annotation procedure followed our institution's ethical research guidelines and received approval from the relevant ethics oversight process.

## C    LIMITATIONS

While deceptive humor represents a novel and underexplored phenomenon, the Deceptive Humor Dataset (DHD) nevertheless has notable limitations that we must acknowledge:

- **Reliance on synthetic data:** The primary limitation of DHD is that it is synthetically generated. This was a deliberate and necessary choice, as large-scale, high-quality organic deceptive humor is extremely difficult to collect. In pilot studies, candidate examples from social media were often ambiguous, and even expert annotators disagreed on labels. We believe this challenge arises from the intrinsic ambiguity and elusiveness of deceptive humor, rather than from any shortcoming in our methodology.

- **Cultural and geographical bias:** The current dataset primarily covers Indian fake news, with widely debunked claims collected from verified fact-checking websites such as Alt News and FactChecker. As a result, DHD may not generalize globally and contains inherent cultural biases. We acknowledge this limitation and plan to extend future work to include more diverse and international sources.

- **Limitations in capturing humor and cultural nuances:** Although we employ a careful generation pipeline with human-in-the-loop refinement, DHD may not fully replicate human-level humor and cultural nuances. As Table 9 shows, for low-resource languages such as Telugu, Hindi, Kannada, and Tamil, preserving culturally aligned humor remains challenging. This reflects an open research problem in aligning LLM-generated content with cultural context.

- **Model-induced bias from using ChatGPT-4o:** Because all synthetic samples were generated with ChatGPT-4o, DHD may inherit stylistic or distributional biases from this model. To mitigate this, we applied human-in-the-loop filtering, multilingual quality checks, and cross-lingual consistency validation. Notably, using a single generator also provides a controlled benchmark setting by avoiding cross-model variation that could confound evaluation.

Accordingly, we position DHD as a controlled and scalable testbed rather than a perfect reflection of the real world. It provides clear ground truth with a known set of fabricated claims, enabling rigorous, apples-to-apples comparison of model capabilities. This approach aligns with prior uses of synthetic data to create foundational benchmarks for otherwise intractable problems (Wang et al., 2022; Liu et al., 2024). Our human-in-the-loop process and evaluation results (see Table 8, Table 9) indicate that the synthetic comments exhibit consistent and recognizable patterns of deceptive humor.

Nonetheless, we acknowledge the "sim-to-real" gap: models trained on DHD may struggle to generalize to organic deceptive humor (Shrivastava et al., 2017; Dong et al., 2022). To systematically bridge this gap, future work will introduce DHD-HARD to test model robustness against human-authored, unstructured variations of the controlled patterns established in DHD. Following best practices for bridging synthetic and real distributions (Kang et al., 2024), evaluating models on DHD-HARD will allow us to quantify and mitigate this gap, establishing a principled agenda for future research in this nascent field.

# D  ABLATION STUDY

Table 6: Ablation study results for DH-MTL. Each row represents a different model configuration with one or both dynamic loss weights removed. Metrics include Accuracy (Acc), Macro F1 (MacF1), and Weighted F1 (WgtF1) for both tasks, and Pearson Correlation (Pear) for Satire Level.

| Model Configuration | Satire Level | | | | Humor Attribute | | |
|---|---|---|---|---|---|---|---|
| | Acc | MacF1 | WgtF1 | Pear | Acc | MacF1 | WgtF1 |
| **Proposed Work (Full DH-MTL)** | | | | | | | |
| DH-MTL (Original) | **46.22** | **42.89** | **43.53** | **33.57** | **36.56** | **31.82** | **34.00** |
| **Ablated Models** | | | | | | | |
| DH-MTL fixed satire weight ($w_{sat}$) | 42.44 | 38.15 | 39.24 | 20.98 | 33.44 | 26.99 | 29.61 |
| DH-MTL fixed humor weight ($w_{hum}$) | 42.89 | 38.95 | 39.92 | 26.54 | 34.11 | 29.69 | 31.87 |
| DH-MTL fixed weights ($w_{sat}, w_{hum}$) | 42.00 | 36.26 | 37.64 | 21.24 | 34.33 | 27.52 | 30.15 |

The ablation study is designed to assess the importance of dynamic loss weighting in the DH-MTL framework and its impact on modeling deceptive humor. Dynamic weighting allows the model to adaptively balance the contributions of the Satire Level and Humor Attribute tasks during training, preventing one task from dominating the other and ensuring both tasks are learned effectively. To quantify this effect, we evaluated three ablated configurations:

1. **Fixed Satire Weight** ($w_{sat} = 1$): In this configuration, the weight for the Satire Level loss is held constant as a scalar, while the Humor Attribute weight remains learnable and updates according to the data. This allows the model to focus on dynamically adjusting the contribution of humor-related features while keeping satire intensity influence static. Comparing this configuration to the full DH-MTL model helps us understand how much the model relies on adaptive weighting for the Satire Level task specifically.

2. **Fixed Humor Weight** ($w_{hum} = 1$): Here, the weight for the Humor Attribute loss is fixed, and the Satire Level weight is allowed to adapt during training. This setup evaluates the impact of removing adaptive control from the humor-related task while retaining flexibility for modeling satire intensity. It demonstrates the benefit of letting the Satire Level task adjust dynamically relative to Humor Attribute, highlighting the interplay between the two tasks in multi-task learning.

3. **Fixed Weights** ($w_{\textbf{sat}} = w_{\textbf{hum}} = 1$): In this configuration, both task weights are held constant, effectively removing the dynamic balancing mechanism entirely. This extreme scenario shows the consequences of treating both tasks equally at all times, without accounting for varying difficulty, learning pace, or task interactions. The resulting drop in performance underscores the necessity of dynamic loss weighting for achieving optimal task synergy.

Table 6 presents the results for all three ablated configurations alongside the full DH-MTL model. The table clearly shows that:

- The full DH-MTL model with dynamically learned weights consistently achieves the best performance across both Satire Level and Humor Attribute tasks, particularly in Pearson correlation for Satire Level, highlighting its ability to preserve ordinal intensity structure.

- Fixing the Satire weight leads to a significant drop in Satire Level performance, confirming that adaptive weighting is critical for capturing intensity differences accurately. Humor Attribute metrics also decrease, but less severely, indicating partial resilience due to the still-dynamic humor weight.

- Fixing the Humor weight primarily impacts Humor Attribute metrics, while Satire Level performance remains relatively better due to the dynamic adjustment of its weight. This shows that task-specific weighting provides the necessary flexibility for learning task interactions.

- Fixing both weights results in the largest overall performance drop, emphasizing that the dynamic balancing mechanism is essential for jointly modeling satire intensity and humor style. Without it, the model struggles to maintain class balance and to differentiate fine-grained humor attributes.

Overall, this detailed ablation study demonstrates that adaptive loss weighting is not merely an auxiliary feature, but a core component of DH-MTL. It enables the framework to allocate learning capacity effectively between tasks, maintain class coverage, capture fine-grained distinctions, and respect the ordinal nature of Satire Level, thereby validating the design choice and motivating its use in deceptive humor detection.

## D.1 IMPORTANCE OF EACH COMPONENT IN DH-MTL

The proposed DH-MTL framework integrates several key components that collectively enhance multi-task learning for deceptive humor detection. Ablation studies (Table 6) highlight their individual contributions:

- **Dynamic Loss Weighting:** Adapts the relative importance of satire and humor tasks during training. This prevents one task from dominating and ensures efficient learning. Removing this mechanism significantly degrades Satire Level performance, particularly Pearson correlation, demonstrating its critical role in modeling ordinal intensity.

- **Ordinal Regression for Satire:** Captures the inherent ordering of satire intensities, enabling the model to distinguish subtle transitions from low to high satire. This yields higher accuracy and correlation than flat classification approaches.

- **Focal Loss for Humor:** Mitigates class imbalance by emphasizing underrepresented humor styles. This improves Macro and Weighted F1 scores compared to standard cross-entropy, allowing better recognition of rare yet meaningful stylistic cues.

- **Two-Stage Training:** Stage 1 (domain adaptation) aligns the encoder with deceptive humor patterns, while Stage 2 (task-specific fine-tuning with gradual unfreezing) refines task specialization. This strategy produces richer representations, boosting performance across both tasks.

Collectively, these components enable the proposed DH-MTL two-stage training framework to outperform traditional approaches and ablated variants, achieving superior metrics by explicitly modeling task interactions and ordinal structure, while leveraging domain adaptation followed by task-specific fine-tuning.

# E    PARAMETER SENSITIVITY ANALYSIS (EXTENDED)

To understand the robustness of DH-MTL, we analyze the sensitivity of the model to key training hyperparameters: batch size and learning rate. We evaluate how different configurations affect both Satire Level and Humor Attribute performance, with results summarized in Table 7.

Table 7: Parameter Sensitivity Analysis for DH-MTL. Each row shows model performance under different batch sizes and learning rates. Metrics include Accuracy (Acc), Macro F1 (MacF1), Weighted F1 (WgtF1) for both tasks, and Pearson Correlation (Pear) for Satire Level.

| Hyperparameter Set | Satire Level | | | | Humor Attribute | | |
|---|---|---|---|---|---|---|---|
| | Acc | MacF1 | WgtF1 | Pear | Acc | MacF1 | WgtF1 |
| Batch=16, LR=2e-10 (Original) | 46.22 | 42.89 | 43.53 | 33.57 | 36.56 | 31.82 | 34.00 |
| Batch=64, LR=2e-10 | 44.56 | 41.02 | 41.92 | 27.68 | 33.67 | 26.16 | 28.90 |
| Batch=16, LR=1e-10 | 42.33 | 39.35 | 40.09 | 24.92 | 33.44 | 25.12 | 28.20 |

The analysis shows that the proposed DH-MTL framework is largely stable across reasonable hyperparameter variations. While the original setting (Batch=16, LR=2e-10) yields the best performance, increasing the batch size or lowering the learning rate slightly reduces accuracy and Pearson correlation for Satire Level, as well as the classification metrics for Humor Attribute. Importantly, the trends confirm that DH-MTL consistently maintains a balanced treatment of classes and preserves the ordinal structure of Satire Level, highlighting that the model is robust and not overly sensitive to moderate changes in training configuration. This stability provides confidence in the practical applicability of the framework across diverse experimental settings.

# F    WHY DECEPTIVE HUMOR IS DANGEROUS

Deceptive humor often appears amusing or harmless at first glance, but closer examination reveals that it can carry false or misleading information along with the comedic content. While such content may provoke laughter, it can simultaneously influence beliefs, reinforce stereotypes, and spread misinformation. Humans are particularly susceptible to deceptive humor because the comedic framing can lower critical scrutiny, making audiences more likely to accept false claims as true unless they are certain of the facts. This dual nature makes deceptive humor a subtle yet powerful vehicle for misinformation that can have psychological, social, and cultural consequences. When viewed through the lens of modern AI systems, an additional concern emerges: if deceptive humor is present in training data without explicit labeling, language models may inadvertently internalize and replicate the same misleading humorous tone, making downstream detection and moderation of such content significantly more difficult.

For instance, we reference the fake claim: ***"Ch\*na is spreading COVID as a bioweapon."*** This example is included not to endorse the statement, but to demonstrate how false claims, when framed in humorous or exaggerated ways, can propagate widely and cause real-world harm. Fact-checking sources[3] have debunked this claim, yet it continues to circulate online. Beyond spreading health-related misinformation, such content can also target specific countries or communities, in this case potentially disturbing individuals who are from, support, or are associated with Ch\*na. This illustrates how deceptive humor can simultaneously misinform, provoke social tension, and amplify xenophobic or geopolitical narratives.

Understanding the propagation of such content is crucial for research in online misinformation and social media safety. By analyzing deceptive humor, we can develop better detection methods, design datasets that account for both the comedic and misleading aspects of content, and ultimately mitigate its harmful effects without censoring legitimate humor. Our study emphasizes the importance of responsible handling and careful annotation of politically or socially sensitive examples to ensure ethical and scientifically sound research.

---

[3]For a comprehensive debunking of the "COVID-19 as a bioweapon" claim, see Snopes and Reuters.

## G  EMPIRICAL EVIDENCE OF HUMOR CONTRIBUTING TO MISINFORMATION PROPAGATION

To validate the relevance of deceptive humor as a meaningful research direction, we review empirical evidence showing how humor can influence the perception, spread, and believability of false claims. Humor plays a prominent role in modern social media ecosystems, where lightweight, shareable formats such as memes, satire, and humorous commentary can accelerate information diffusion.

Humorous framing can reduce epistemic vigilance and shift attention away from factual evaluation. Behavioral and eye-tracking studies indicate that humor modulates attention allocation, lowering perceived seriousness of claims and weakening analytical scrutiny (Yeo & McKasy, 2021). Consequently, audiences may accept or share false claims more readily when they are embedded in humorous content.

Humor also enhances social transmission. Memes, ironic posts, and pithy humorous quips tend to be more widely shared due to social-signaling motives, in-group bonding, and heightened emotional engagement. As a result, fabricated claims packaged in humorous formats propagate further than non-humorous content, even when their credibility is low (Rodríguez-Ferrándiz et al., 2021; Flecha Ortiz et al., 2021).

A recurring challenge highlighted in prior work is the ambiguity between satire and literal misinformation. Audiences lacking contextual cues may fail to distinguish parody from deception, allowing false claims to circulate in humorous forms without being clearly identified as misinformation (Boukes & Hameleers, 2023). This supports operationalizing satire as a graded signal rather than a binary label.

Importantly, humor can play a dual role: it may correct misinformation, but it can also reinforce false beliefs depending on the audience and context (Yeo & McKasy, 2021; Muhammed T & Mathew, 2022). Humorous interventions can backfire, strengthening identity-driven beliefs or inadvertently normalizing false claims. This dual effect underscores the nuanced role of humor in shaping information ecosystems.

**Relevance to DHD:** These findings directly inform the design of the Deceptive Humor Dataset. By holding the presence of a fabricated claim constant and annotating how humor is expressed, via Satire Level and Humor Attributes (or Humor stylistic attributes), we capture the mechanisms through which deceptive humor shapes believability and detectability. Satire Level reflects gradations of subtlety or exaggeration, while Humor Attributes encode rhetorical strategies such as irony, absurdity, social commentary, wordplay, and dark humor. Collectively, these annotations enable detailed analysis of how humorous framing can normalize, obscure, or amplify false claims, providing structured supervision for models to study the pathways through which deceptive humor influences audiences.

## H  REAL-WORLD DECEPTIVE HUMOR ANALYSIS

In real-world settings, especially during crises, deceptive humor is a powerful tool in the information ecosystem. During the COVID-19 pandemic, misinformation surged, not only as serious rumors but also as humorous, satirical content. For instance, memes anthropomorphizing the virus ("Corona-chan") used gallows humor to propagate conspiracy theories about its origins, subtly embedding falsehoods in comedic form[4]. Empirical research also shows that health-related misinformation spread more rapidly than other content[5].

More recently, geopolitical tension between Ind*a and Pak*stan has given rise to wartime disinformation campaigns amplified in humorous or satirical formats. A flurry of fake videos, manipulated images, and exaggerated stories circulated online, including false claims of missile strikes, drone attacks, and nuclear escalation[6]. For example, some social media users joked about "nuclear radiation spreading" after purported attacks, using humor to amplify fear and speculation[7].

---

[4]Corona-chan Wikipedia
[5]COVID-19 misinformation spread study
[6]India Today report on disinformation
[7]The Tribune report

These humorous distortions were not just grassroots memes. Observers noted a coordinated information campaign: a think-tank analysis found that many disinformation posts came from influential accounts with high visibility[8]. Media fact-checkers, such as the Press Information Bureau (PIB), also debunked several high-profile hoaxes, such as a WhatsApp message alleging "Operation Sindoor," falsely claiming imminent conflict operations[9]. Meanwhile, social media users in both countries produced satirical content that blurred the line between mockery and propaganda[10].

This real-world usage demonstrates three critical points:

1. **Deceptive humor is both topical and timely**, emerging strongly during periods of crisis or tension.

2. **Humor helps misinformation spread** by reducing skepticism and embedding disinformation in culturally resonant or absurd formats.

3. **The framing of a false claim through humor** is not merely decorative, humor changes how people perceive and share misinformation, making it harder to combat with standard fact-checking methods.

These patterns strongly motivate our DHD dataset: by modeling how humor frames deception (through Satire Level and Humor Attribute), we aim to capture the *stylistic mechanisms* that enable false narratives to thrive in real-world settings. Understanding these mechanisms is essential for building systems that can detect not only outright falsehoods but also the *humorous packaging* that makes them socially acceptable and shareable.

## I    HUMAN-IN-THE-LOOP DATASET GENERATION

**Motivation:**
To ensure high-quality generation of deceptive humor comments across multiple languages and code-mixed variants, we employed a human-in-the-loop (HITL) workflow. In this setup, content was batch-generated by ChatGPT. It was then rigorously reviewed by a faculty member and a PhD researcher specializing in computational humor, as well as three postgraduate students (with informed consent), who ensured linguistic accuracy, label correctness, and coherence. Such HITL supervision of synthetic generation has been shown to significantly improve content quality and reduce errors. For example, provenance-based HITL frameworks like INSPECTOR demonstrate that incorporating human review into AI-generated corpora leads to 3-4× more accurate labels compared to unsupervised generation (Kang et al., 2024). This structured oversight ensured that the final DHD dataset met high standards of quality (see subsection I.2 for the structured generation prompt).

### I.1    WORKFLOW OVERVIEW

The HITL process was conducted as follows:

1. **Batch Generation:** Comments were generated by the model in small batches of 100 records per iteration, using a prompt that included the fake news statement as input.

2. **Manual Evaluation:** Each batch of 100 generated samples was carefully reviewed to ensure quality. The checks included:
   - Verifying correctness of assigned labels.
   - Assessing grammatical structure and readability.
   - Ensuring meaningfulness of the comment and removing nonsensical or irrelevant outputs.

3. **Iterative Feedback:** Any detected errors (label mismatches, grammatical issues, or meaningless comments) were fed back to the model with instructions to correct them. This loop continued until the batch met quality standards.

---

[8]Economic Times on social media campaigns
[9]Moneycontrol fact-check
[10]The Guardian report

4. **Final Compilation:** Once each batch of 100 comments passed quality checks, it was added to the growing DHD dataset. Repeating this process iteratively over multiple batches produced the final dataset of 9,000 multilingual and code-mixed deceptive humor comments.

This careful dataset generation pipeline was designed to closely mimic real-world data distributions, producing machine-generated text that resembles human-authored deceptive humor. We believe that incorporating iterative human-in-the-loop review substantially improves the quality and reliability of the dataset compared to standard prompting methods. While this process enhances linguistic correctness and label accuracy, we acknowledge that it may not fully capture all patterns and subtleties of naturally occurring deceptive humor in the wild.

## I.2 PROMPT USED FOR DATASET GENERATION

The following prompt was used to guide model generation:

---

**Dataset Generation Prompt**

**\<System\>**

You are tasked with generating humorous comments in one of the following languages: English (En), Telugu (Te), Hindi (Hi), Kannada (Ka), Tamil (Ta), and their code-mixes (Te-En, Hi-En, Ka-En, Ta-En).

Each generated comment must include the following labels:
**Satire Level**:
- Low Satire: Subtle humor, lightly satirical.
- Moderate Satire: Evident humor, incorporating exaggeration or sarcasm.
- High Satire: Strongly exaggerated or overtly satirical.

**Humor Attribute**:
- Irony: Intended meaning contrasts with literal meaning.
- Absurdity: Exaggeration or illogical scenarios.
- Social Commentary: Critiques societal or cultural issues.
- Dark Humor: Morbid, taboo, or controversial topics.
- Wordplay: Puns, double meanings, or clever linguistic constructs.

Ensure comments are complex, human-like, and reflect various humor types.
**\</System\>**

**\<Hypothesis\>**
Generate a humorous comment based on the following fake claim:
**\</Hypothesis\>**

**\<User\>**
Please generate a humorous comment based on the fake claim: $<$FAKE_CLAIM$>$
**\</User\>**

---

## I.3 NOTES ON HITL QUALITY CONTROL

- Random sampling and author review ensured consistency and quality across batches.

- Iterative feedback allowed the model to correct errors, improving label accuracy and comment naturalness.

- The process ensured coverage of multiple languages, code-mixed variants, and diverse humor types, resulting in a high-quality, multilingual deceptive humor dataset suitable for LLM evaluation.

## J  ROLE OF SYNTHETIC DATA IN ADVANCING THE AI SYSTEMS

The proposed DHD is synthetically generated using the ChatGPT-4o model. A common critique of synthetic data is that PLMs struggle to capture patterns representative of human-generated text. While this concern has some merit, it is important to recognize that human annotations themselves are influenced by inherent biases shaped by individual mental models (Gautam et al., 2024). The role of synthetic data in AI research has grown substantially, with top institutions like Hugging Face and various companies actively developing synthetic data generators[11] to support this effort. Notably, the Phi-4 model, a SOTA open model, incorporates synthetic data as a core component of its training regimen, underscoring its practical value in advancing AI capabilities.

Recent work across leading AI research venues further validates synthetic data's critical role in improving model generalization, addressing data scarcity, and mitigating annotation biases. For instance, Google DeepMind's comprehensive study outlines best practices and challenges in synthetic data generation, highlighting its potential to enhance model robustness and fairness (Liu et al., 2024). In multimodal learning, synthetic data has been demonstrated to boost unsupervised visual representation learning by generating effective training samples and improving data efficiency (Wu et al., 2023). Additionally, synthetic data-driven self-training methods have shown promise in low-resource natural language processing tasks such as relation extraction, effectively overcoming domain adaptation challenges (Xu et al., 2023). These advancements position synthetic data not merely as a workaround for limited human annotations but as a transformative tool that drives innovation and broader applicability in AI. In this light, synthetic data provides an essential foundation for our work on Deceptive Humor detection and enables future research progress in this potentially emerging domain.

**Reference Links**

**Humor Types:**

- **Irony:** Irony Wikipedia
- **Absurdity:** Absurdity Wikipedia
- **Social Commentary:** Social Commentary Wikipedia
- **Dark Humor:** Dark Humor Wikipedia
- **Wordplay:** Wordplay Wikipedia

**Fact-Checking Sources:**

- **AltNews:** https://www.altnews.in/
- **Boom FactCheck:** https://www.boomlive.in/fact-check
- **FactChecker:** https://www.factchecker.in/fact-check
- **FACTLY:** https://factly.in/

## K  HUMAN EVALUATION (EXTENDED)

In this section we discuss the extended version of Human Evaluation. Our human evaluation setup was designed not only to validate labeling reliability but also to examine how deceptive humor manifests across languages with distinct cultural, morphological, and code-mixed characteristics. Table 8 reports pairwise and multi-annotator agreement for both Satire Level and Humor Attribute. While English shows the strongest alignment, the Indic languages exhibit systematic variability: disagreements frequently arise in borderline cases where humor, sarcasm, and misinformation intertwine in ways that depend heavily on socio-political context. This pattern is consistent across all annotator pairs and is further reflected in the weighted Kappa values, suggesting that divergences are not random noise but stem from genuine ambiguity in interpreting the satirical intensity of misinformative humor. Notably, Fleiss' Kappa remains in a stable moderate range across languages, indicating that despite individual variability, annotators converge on shared judgment patterns at the dataset scale.

---

[11]https://huggingface.co/blog/synthetic-data-generator

These agreement scores therefore provide a realistic picture of the interpretive uncertainty inherent in deceptive humor, rather than an artifact of annotation quality alone.

To further characterize the fidelity and naturalness of the synthetic data, we performed a structured quality assessment covering Readability, Claim-Graspability, and Cultural Nuance (see Table 9). The cross-lingual trends reveal a clear stratification: English consistently achieves high scores, while morphologically rich Indic languages show drops that correlate with two recurring phenomena: (i) narrative drift, where the humorous surface slightly obscures the underlying false claim, and (ii) cultural mismatches, where the generative model defaults to humor tropes not commonly used by native speakers. These deviations are especially prevalent in Kannada and Tamil, suggesting that low-resource pretraining signals and limited exposure to region-specific humor styles weaken the model's ability to generate culturally grounded deception. Importantly, even with these variations, Claim-Graspability remains above threshold across all languages, indicating that the core misinformative narrative remains interpretable to human evaluators. Taken together, the agreement and quality analyses illustrate the linguistic and cultural gradients that shape deceptive humor generation, providing valuable diagnostic insights for future cross-lingual robustness and controllability efforts.

Table 8: Inter-annotator agreement scores for Satire Level and Humor Attribute. Top: Satire Level (Unwt K = unweighted Cohen's Kappa; Wtd K = weighted Cohen's Kappa; F-K = Fleiss' Kappa across all three annotators). Bottom: Humor Attribute (Unwt K = unweighted Cohen's Kappa; Wtd K = not applicable; F-K = Fleiss' Kappa). (*) indicates Indic languages along with their code-mixed variants.

| Lang | Mac vs Ann1 | | Mac vs Ann2 | | Ann1 vs Ann2 | | F-K |
|------|-------------|---|-------------|---|--------------|---|-----|
| | Unwt K | Wtd K | Unwt K | Wtd K | Unwt K | Wtd K | |
| **Satire Level** | | | | | | | |
| English | 52.75 | 61.93 | 65.24 | 66.70 | 65.49 | 69.40 | 63.21 |
| Telugu* | 45.06 | 58.43 | 63.27 | 64.96 | 56.13 | 59.65 | 55.26 |
| Hindi* | 53.41 | 60.69 | 70.43 | 77.66 | 65.71 | 69.00 | 62.86 |
| Kannada* | 49.86 | 61.92 | 68.56 | 77.19 | 63.64 | 64.64 | 60.37 |
| Tamil* | 45.97 | 52.66 | 64.12 | 67.80 | 61.47 | 65.55 | 58.20 |
| **Overall** | 51.87 | 62.97 | 67.65 | 73.30 | 67.25 | 68.51 | 62.06 |

| Lang | Mac vs Ann1 | | Mac vs Ann2 | | Ann1 vs Ann2 | | F-K |
|------|-------------|---|-------------|---|--------------|---|-----|
| | Unwt K | Wtd K | Unwt K | Wtd K | Unwt K | Wtd K | |
| **Humor Attribute** | | | | | | | |
| English | 65.62 | – | 69.82 | – | 76.05 | – | 70.44 |
| Telugu* | 53.37 | – | 66.52 | – | 59.91 | – | 59.88 |
| Hindi* | 56.14 | – | 67.03 | – | 65.01 | – | 62.64 |
| Kannada* | 54.55 | – | 71.39 | – | 62.91 | – | 62.89 |
| Tamil* | 50.91 | – | 68.02 | – | 67.20 | – | 61.95 |
| **Overall** | 55.64 | – | 68.64 | – | 65.74 | – | 63.30 |

Table 9: Quality Assessment for Deceptive Humor Data (scale of 1–10; 1 = low, 10 = high).

| Lang | Readability | Claim-Graspability | Cultural Nuance |
|------|-------------|--------------------|-----------------|
| English | 9.00 | 7.80 | 8.20 |
| Telugu* | 7.90 | 7.60 | 7.10 |
| Hindi* | 7.70 | 7.20 | 7.20 |
| Kannada* | 7.10 | 6.30 | 6.20 |
| Tamil* | 7.30 | 6.50 | 5.90 |

## L ERROR ANALYSIS

This section provides a deep dive into instances where the model misclassified Satire Level and Humor Attribute. We analyze the original labels, predictions, and the potential reasons behind these errors. Such an analysis uncovers recurring challenges, particularly in detecting indirect satire, cultural nuances, and subtle humor constructs.

**Satire Level:** We focus on cases where comments originally labeled as High Satire (level 3) were predicted as Low Satire (level 1). These examples highlight the difficulty models face when satire is implied, indirect, or culturally embedded.

---

**Example 1: Satire Level**

**Comment:** Are temples now just stepping stones for interfaith harmony campaigns?
**Original Label:** 3 (High Satire)
**Predicted:** 1 (Low Satire)
**Possible model interpretation:** The model likely focused on the literal meaning of the words, missing the subtle critique of interfaith campaigns.
**Conclusion:** Indirect satire that relies on cultural context or social commentary is particularly challenging for models.

---

**Example 2: Satire Level**

**Comment:** Seems like nud*ty is the new immunity booster.
**Original Label:** 3 (High Satire)
**Predicted:** 1 (Low Satire)
**Possible model interpretation:** The exaggerated statement was subtle and implied, so the model misjudged the level of satire.
**Conclusion:** Models tend to underestimate satire when it is indirect or lightly veiled rather than overt.

---

**Example 3: Satire Level**

**Comment:** Sure, cutting taxes for billionaires will totally fix climate change.
**Original Label:** 3 (High Satire)
**Predicted:** 2 (Medium Satire)
**Possible model interpretation:** The model recognized some level of exaggeration but failed to fully capture the sarcastic critique of policy decisions.
**Conclusion:** Even explicit sarcasm mixed with societal critique can be partially misclassified if the context is nuanced.

---

**Humor Attribute:** Next, we examine examples where the model confused one humor type for another, highlighting challenges in detecting nuanced styles such as irony, wordplay, absurdity, and satirical exaggeration.

---

**Example 1: Humor Attribute**

**Comment:** Ch*na's virus: the only war fought with sweatpants and Wi-Fi!
**Original Label:** Absurdity
**Predicted:** Irony
**Possible model interpretation:** The contrast between "war" and "sweatpants/Wi-Fi" may have appeared ironic rather than absurd due to the unexpected comparison.
**Conclusion:** Extreme exaggeration can be misread as irony; distinguishing between humor types requires understanding the intended incongruity.

---

---

**Example 2: Humor Attribute**

**Comment:** So the gas cylinder decided to skip being cooked on and went straight for a track record?
**Original Label:** Wordplay
**Predicted:** Irony
**Possible model interpretation:** The pun involving "track record" may have been interpreted as ironic commentary rather than playful language.
**Conclusion:** Clever wordplay is often misclassified as irony, reflecting the difficulty of detecting subtle linguistic constructions.

---

**Example 3: Humor Attribute**

**Comment:** The new vaccination program is so good, they've started giving out free loyalty cards.
**Original Label:** Absurdity
**Predicted:** Social Commentary
**Possible model interpretation:** The model interpreted the absurd exaggeration of "loyalty cards" as a critique of public health campaigns or as commentary on consumer culture and how it's being applied to public services. This is a common conflation, as social commentary often uses absurdity to make its point.
**Conclusion:** Models can confuse humor that relies on extreme exaggeration (Absurdity) with Social Commentary, highlighting a challenge in distinguishing between purely fantastical humor and humor with a pointed, satirical target.

---

**Key Insights:**

- Indirect and culturally embedded satire is frequently underestimated by models.

- Exaggeration vs. irony vs. absurdity distinctions are subtle and often confused, indicating the need for stronger contextual and pragmatic understanding.

- Wordplay and puns are particularly challenging, as surface-level semantic similarity may mislead models.

- Overall, error patterns highlight that deceptive humor requires not just lexical understanding, but reasoning over intent, cultural nuance, and layered meanings.

## M  DATASET DESCRIPTION SUMMARY

Table 10: DHD distribution across train, validation, and test sets.

| Statistic | Train | Validation | Test |
|---|---|---|---|
| Total Samples | 7,200 | 900 | 900 |
| **Satire Level Distribution** | | | |
| Low Satire | 2,080 | 276 | 367 |
| Moderate Satire | 3,138 | 382 | 227 |
| High Satire | 1,982 | 242 | 306 |
| **Humor Attribute Distribution** | | | |
| Irony | 2,200 | 282 | 259 |
| Absurdity | 1,661 | 180 | 248 |
| Social Commentary | 1,215 | 155 | 106 |
| Dark Humor | 1,089 | 136 | 160 |
| Wordplay | 1,036 | 147 | 127 |

