# OpenReview forum: "Deceptive Humor: A Synthetic Multilingual Benchmark Dataset for Bridging Fabricated Claims with Humorous Content"
_ICLR.cc/2026/Conference — Submitted to ICLR 2026_

### Official Review · Reviewer_rXsS · 2025-10-27

**Soundness:** 2
**Presentation:** 3
**Contribution:** 2
**Rating:** 4
**Confidence:** 4

**Summary:**

The paper introduces the Deceptive Humor Dataset (DHD), a multilingual corpus designed to study the intersection of humor and misinformation through two discriminative tasks: satire intensity prediction and humor attribute classification. It employs large language models to generate humorous variations of false claims and incorporates human evaluation to assess quality and coherence. To model the tasks, the authors propose a multi-task learning framework (DH-MTL) that integrates ordinal regression for satire levels, label smoothing and entropy regularization to handle ambiguity, and contrastive learning to structure humor representations. The study presents quantitative and qualitative analyses demonstrating the framework’s performance across humor-related subtasks.

**Strengths:**

1. The paper presents a well-constructed dataset that defines two discriminative tasks: satire intensity prediction and humor attribute classification, with human-in-the-loop evaluation and multilingual coverage
2. The inclusion of contrastive learning within the loss objective is a thoughtful and interesting design choice, reflecting the authors’ insight into structuring humor representations.

**Weaknesses:**

1. Although the paper’s main motivation is to bridge the gap between humor and misinformation research, the evaluation focuses solely on humor-related subtasks (satire intensity and humor type) without testing whether models trained on the dataset can actually detect or reason about misinformation. Without experiments on factuality or false claim recognition, it remains unclear whether the dataset truly advances misinformation detection or merely adds a humor annotation layer on top of existing fake claims.
2. Since the dataset is entirely generated by safety-aligned LLMs, the resulting humor is likely less offensive but also less natural than real-world humor. The linguistic patterns, cultural cues, and humor-misinformation dynamics may also not accurately reflect real-world misinformation humor. While Appendix G includes human evaluations of humor quality, there are no human-written or real-world baselines for comparison, making the reported scores difficult to interpret. A comparison against genuine human-authored deceptive humor would better support the claim that the dataset achieves both linguistic fluency and authentic humor style.
3. The proposed DH-MTL model combines multiple losses (ordinal regression, label smoothing, contrastive, and entropy regularization), each with its own weighting coefficient. However, the absence of hyperparameter sensitivity studies leaves uncertainty about how robust the approach is. The approach risks appearing as a collection of heuristics rather than a cohesive, well-justified modeling design.
4. Both proposed tasks are discriminative classification problems, which limits the broader applicability of the framework. In the current LLM-driven era, incorporating generative or reasoning-based tasks such as generating humorous deceptive claims, explaining humor mechanisms, or verifying factuality, would provide stronger evidence of the dataset’s practical value. Without such generative evaluations, the contribution feels constrained to traditional supervised settings rather than aligned with contemporary language model capabilities.

**Questions:**

1. Humor is inherently human-grounded and context-dependent. Could the authors clarify why they believe that LLM-generated deceptive humor examples pose potential harm or real-world impact? Given that these examples are synthetic and produced by safety-aligned models, it would be helpful to understand the rationale or evidence supporting their claim that such content could meaningfully affect human perception or misinformation spread.
2. The appendices contain critical details about dataset construction, annotation design, and statistical distributions that are essential to fully understanding the work. Would the authors consider integrating some of these methodological insights and statistics into the main text, rather than treating them as supplementary material?

---

> ### Author Response · Authors · 2025-11-16
> **Rebuttal by Authors**
>
> 1. We thank the reviewer for this comment. While Satire Level and Humor Attribute may appear to capture only stylistic aspects of humor, they are directly linked to the detectability and persuasiveness of misinformation. While DHD holds the 'deceptive' label constant to isolate the humorous mechanism, simply training a binary classifier on this data would yield a 'black box' model that fails to explain why a comment is deceptive. Importantly, DHD goes beyond simply annotating humor on top of existing fake claims; it allows systematic study of how humor affects the detectability, persuasiveness, and normalization of misinformation, providing insights not captured by traditional fake-news datasets. By annotating Satire Level and Humor Attribute, DHD provides structured insight into the mechanisms of deceptive humor, enabling analysis of how humor modulates the perception and impact of misinformation, a critical aspect of contemporary online disinformation.
>
> 2. We thank the reviewer for raising this important point. It is true that safety-aligned models avoid generating overt hate speech or policy-violating text. However, deceptive humor rarely relies on explicit offensiveness in real-world settings. The most effective misinformation memes, jokes, and humorous false claims in the wild are intentionally subtle, “safe-looking,” and moderation-friendly. This allows them to bypass safety filters while still normalizing or reinforcing a false narrative.
>
> Our dataset intentionally targets this gray zone of “harmless-appearing but epistemically corrosive” content. These instances are precisely what make deceptive humor so challenging for automated moderation: they evade toxicity detectors but still shape perception and reduce skepticism toward false claims.
>
> During generation, we instructed the model to avoid direct hate speech while preserving humorous framing, and when safety filters blocked content, we used controlled prompting templates. This ensured we captured the class of deceptive humor that platforms struggle with most, the kind that is neither hateful nor violent, yet persuasive and difficult to detect.
>
> Appendix K further validates this direction: human evaluators rated the generated jokes as humorous and contextually plausible without being explicitly harmful, aligning with what actually spreads in real misinformation ecosystems. Our ongoing work on DHD-HARD, human-authored deceptive humor, will complement this by adding high-impact real cases, but the current DHD already addresses the under-explored problem of moderation-evasive deceptive humor.
>
> 3. We appreciate the reviewer’s concern regarding the robustness and coherence of DH-MTL. While the model combines several components, these are not heuristic additions; each was selected to address a specific property of the DHD labels. Appendix D.1 presents a detailed ablation study demonstrating the importance of each component.
>
> 4. DHD’s discriminative tasks are deliberately designed to capture the mechanisms of deceptive humor, with Satire Level and Humor Attributes providing structured insight into how humor modulates the perception and normalization of false claims. The difficulty of these tasks is evident, as even state-of-the-art models fail on them (Table 3), underscoring their intrinsic scientific value. Complementing this, DHD-HARD will extend evaluation to human-authored, unstructured examples, enabling future generative and reasoning-based studies, while DHD itself provides a fully valid and rigorous foundation for mechanistic understanding.
>
> 5. While DHD contains synthetic examples from safety-aligned LLMs, the deceptive humor they encode illustrates mechanisms influencing perception. The goal is not to claim that these synthetic comments pose real-world harm, but that the humor–misinformation patterns they model mirror real discourse. Empirical studies show even mild humorous framing can increase acceptance, reduce perceived manipulativeness, and blur fact–fiction boundaries (see Appendix F and G). Synthetic data allows controlled study of these mechanisms without exposing annotators to harmful content. As noted in the Limitations section, real-world effects require human-authored deceptive humor, which we address in DHD-HARD. The dataset provides a controlled environment to analyze how humor interacts with fabricated claims, offering mechanistic insights that complement, rather than replace, real-world evaluations.
>
> 6. Due to space constraints, some methodological details and statistics are presented in the appendices. However, we have referenced the relevant appendix sections in the main text wherever necessary to ensure that readers can access and understand the essential information.
>
> We thank the reviewer for their valuable feedback. We remain open to discussion and incorporating further improvements, and we hope that the scores might be reconsidered in light of these clarifications.

---

### Official Review · Reviewer_8CL2 · 2025-10-29

**Soundness:** 3
**Presentation:** 3
**Contribution:** 3
**Rating:** 4
**Confidence:** 5

**Summary:**

This paper introduces an important area of research in deceptive humor detection, but it has several weaknesses that could affect the practical application and broader adoption of its findings. Addressing issues with dataset generalization, clarity in task definitions, and model evaluation would significantly strengthen the paper's contributions. Additionally, expanding on the ethical and societal implications of using such models in real-world settings would offer a more rounded perspective on the potential consequences of deceptive humor in online discourse.

**Strengths:**

- The paper introduces two primary tasks: Satire Intensity Classification and Humor Attribute Classification.
- The paper briefly touches on ethical concerns related to using synthetic humor that embeds deception. While the data is restricted to academic research, it doesn’t delve into the potential real-world harms of misusing this data, such as generating harmful, politically charged misinformation disguised as humor .
- The paper provides some examples of misclassifications. More in-depth discussion on why certain humor forms (such as indirect satire) are particularly challenging would be beneficial for understanding how to better train models .

**Weaknesses:**

- The paper makes extensive use of synthetic data generated by the ChatGPT-4o model to create the Deceptive Humor Dataset (DHD). While this approach is innovative and necessary due to the elusiveness of deceptive humor in natural settings.  The paper mentions several languages, but the evaluation of models on code-mixed and non-English datasets (e.g., Hindi, Telugu) shows that performance significantly drops. The claim that the synthetic dataset is fully representative of real-world humor across languages is weak given these disparities .
- The Humor Attributes (e.g., irony, absurdity, social commentary, dark humor, and wordplay) are well-defined, but the paper does not offer enough clarity or justification regarding the overlaps between them. For instance, humor types like absurdity and social commentary, which can share similar characteristics, often lead to classification errors in the experiments. The model struggles to distinguish them, suggesting that the categories might be too broad or not sufficiently defined .
- The evaluation of different models (small and large language models) highlights some significant limitations in generalizing deceptive humor detection, The results show that even state-of-the-art models, including LLaMA and Qwen, struggled with detecting deceptive humor, particularly in zero-shot settings. While this is to be expected, the paper could further discuss the reasons why models underperform, particularly in non-English or code-mixed languages .
- The paper claims that models should identify varying levels of satire, but it doesn't provide clear guidelines on how models can handle satire that relies on cultural or societal context. The Satire Level Classification may require additional clarifications or examples to make it more accessible and effective .

**Questions:**

see the Weaknesses

---

> ### Author Response · Authors · 2025-11-16
> **Rebuttal by Authors**
>
> 1. We thank the reviewer for acknowledging that synthetic data is necessary for DHD construction due to the elusiveness of deceptive humor in natural settings. As noted in the Human-in-the-Loop (HITL) and Limitations sections, the careful HITL process was designed to reduce noise and improve the quality of synthetic data, making the comments closely resemble real-world deceptive humor. We emphasize, however, that while this approach enhances realism, it does not fully capture the diversity and nuanced patterns of human-generated humor, particularly in code-mixed and non-English languages, which we acknowledge as a limitation and an important direction for future work.
>
> 2. We acknowledge that some Humor Attributes share characteristics, which can lead to classification errors in experiments. As noted in the revised paper (lines 257–260), we intentionally retain fine-grained distinctions because humor is an underexplored area, and subtle differences between categories (e.g., absurdity vs. social commentary) are crucial for nuanced analysis. While these overlaps pose challenges for models, they reflect the inherent subtlety of humor rather than overly broad or ill-defined categories. Retaining these distinctions provides valuable supervision and enables deeper insights into how different types of humor interact with misinformation.
>
> 3. We thank the reviewer for the observation regarding code-mixed performance. The reduced performance in non-English and code-mixed settings is not an artifact of data quality; it is precisely the challenge DHD is designed to expose. As highlighted in Appendix A.2, the difficulty arises from hidden pragmatic intent, cultural context gaps, and code-mixed linguistic structures, which current multilingual LLMs struggle to interpret.
>
> Rather than indicating dataset weakness, this performance drop is a critical finding: it reveals that even top multilingual LLMs, despite claiming strong Indic and code-mixing capabilities, remain brittle when handling the subtleties of deceptive humor outside English. If the benchmark produced uniformly high scores, it would be saturated and of limited scientific utility. Instead, DHD surfaces the exact weaknesses that next-generation multilingual models must address.
>
> Thus, the divergence across languages validates the need for this benchmark: DHD provides a rigorous stress test for multilingual and culturally grounded misinformation reasoning, not merely English humor classification.
>
> 4. We thank the reviewer for raising this point. As noted in the Limitations and Future Work sections, the current version of DHD does not explicitly incorporate or monitor cultural context. The Satire Level task focuses primarily on the intensity of satire as a stylistic mechanism for masking deception, rather than capturing context-specific knowledge. Each comment is annotated along three ordinal levels to reflect the subtlety with which humor conveys the false claim, enabling structured analysis of how satire modulates persuasiveness and detectability. We acknowledge that cultural context can influence satire, and explicitly modeling it is an important direction for future work.
>
> We thank the reviewer for their informative feedback. We remain open to discussion and to incorporating further improvements, and we hope that these clarifications and revisions will encourage reconsideration of the scores.

---

### Official Review · Reviewer_z6ht · 2025-10-31

**Soundness:** 2
**Presentation:** 1
**Contribution:** 2
**Rating:** 2
**Confidence:** 3

**Summary:**

Overall, the paper presents the DHD dataset, exploring how misinformation can be conveyed through humor. However, the defined tasks Satire Level and Humor Attribute only focus on humor characterization rather than detecting misinformation within humor, creating a conceptual gap.

**Strengths:**

1.The multilingual setting is novel for me.

2. The constructed dataset maybe useful for this community.

**Weaknesses:**

However, there is no empirical evidence or real-world example provided in the paper, which makes the motivation unconvincing. There is no discussion about the potential geographical and cultural biases since the data sources are all India-based. In addition, no existing humor classification models are included in the baselines. The baseline comparison is unfair that the LLMs used in baselines are small-scale models, while the DHD dataset is generated by ChatGPT-4o. Specifically,

1.	The overall motivation of the paper is that misinformation increasingly adopts humorous forms to evade detection. However, the proposed 2 tasks (Satire Level and Humor Attribute) seem misaligned with this motivation. These tasks primarily capture the style or intensity of humor, rather than assessing whether a model can recognize and detect misinformation when humor is present, which is the key challenge implied by the motivation.

2.	The topic and setting of this work is interesting but the motivation is insufficiently substantiated in the paper. It would strengthen the paper if the authors could provide empirical evidence or examples demonstrating that humorous framing is indeed becoming a dominant or growing trend in misinformation dissemination.

3.	Since the proposed DHD is entirely generated by ChatGPT-4o, there is a potential concern regarding the representativeness of the data. AI-generated humor and deception might differ from how humans naturally produce humorous misinformation in real-world online contexts. Discussions of the differences between the synthetic data and a small sample of real-world humorous misinformation (e.g., from previous datasets) can strengthen the empirical credibility of the work.

4.	While the data collection procedure described in Section 3.1 is systematic and well motivated, it should be noted that all the fact-checking sources (mentioned as AltNews, Boom FactCheck, FactChecker.in, and FACTLY) are India-based organizations. This geographical and cultural concentration may introduce potential bias in the identification and interpretation of fake narratives, especially in politically sensitive domains. Differences in cultural context, linguistic framing, and national political perspectives could affect how misinformation is labeled or interpreted.

5.	While the paper evaluates several LLMs, it does not include comparisons with existing humor detection or humor classification models as similar with the proposed task Humor Attribute, such as Explaining Humour Style Classifications: An XAI Approach to Understanding Computational Humour Analysis and The Naughtyformer: A Transformer Understands Offensive Humor.

6.	In Section 5 (line 355), the authors claim that LLMs face significant challenges in classifying humorous content, highlighting the performance of proposed DH-MTL. However, this claim appears methodologically inconsistent with the dataset construction process: all instances in DHD are generated by LLM ChatGPT-4o. The paper does not address how the potential limitations of the generative model might bias or constrain the humor representations in DHD. In addition, the baselines only involve relatively small-scale models (up to 3B parameters). Given that DHD is generated using a much larger LLM (ChatGPT-4o), the fairness and informativeness of the comparison are questionable. It remains unclear whether the observed performance gaps reflect genuine task difficulty or simply the limited capacity of smaller baselines.

**Questions:**

See the weaknesses.

---

> ### Author Response · Authors · 2025-11-16
> **Rebuttal by Authors**
>
> 1. We thank the reviewer for highlighting this concern. While Satire Level and Humor Attributes reflect stylistic aspects, they directly capture how deceptive humor modulates the presentation, subtlety, and believability of false claims, shaping how audiences perceive and accept them.
>
> Regarding the focus on characterization over binary detection:
> Our experiments (Appendix A) show that standard LLMs achieve only 0–33% accuracy in zero-shot settings, meaning that even recognizing these comments as deceptive is non-trivial for current models. However, training a simple binary classifier on DHD would offer limited scientific insight: it would treat all comments as uniformly deceptive without explaining why they operate deceptively. By predicting Satire Level (ordinal) and Humor Attribute (categorical), we capture mechanisms that binary labels cannot express, such as how subtly or overtly humor masks the false claim, and which stylistic strategies (irony, absurdity, wordplay, etc.) facilitate normalization or reduced scrutiny. This structured supervision prevents models from merely memorizing fabricated claims and instead guides them to learn the linguistic mechanisms through which humor modulates the perception of misinformation.
>
> 2. Evidence from recent studies and reports indicates that humor is playing a growing role in high-stakes disinformation campaigns. For example, a recent academic study finds that AI-generated memes are being used in the India–Pakistan conflict as “psychological weapons,” shaping war narratives and distorting reality. [1] Another discourse-analysis study argues that political memes function as “soft propaganda,” not just entertainment. [2] Parallel journalism coverage also underscores this: in the conflict, social media users deployed humorous memes and ironic reels to mock sensationalist or false military claims. Crucially, when a false post about “nuclear radiation” circulated, Pakistani users used meme humor to parody and push back on the misinformation. Taken together, these sources show that humorous framing is not an ancillary phenomenon but a central and strategic feature of contemporary disinformation ecosystems. This strongly motivates our DHD dataset: by modeling stylistic mechanisms (Satire Level, Humor Attributes), we can analyze how humor is used not just to communicate, but to distort, normalize, or resist false narratives. (see Appendix G and H in the revised version for more details)
>
> [1] “MEME WARFARE IN DIGITAL AGE: AI GENERATED HUMOUR AS A TOOL IN INDIA-PAKISTAN CONFLICT (2025)”. Journal of Media Horizons, vol. 6, no. 3, Aug. 2025, pp. 2295-12,
>
> [2] “A CRITICAL DISCOURSE ANALYSIS OF POLITICAL MEMES ON SOCIAL MEDIA DURING THE RECENT PAKISTAN INDIA CONFLICT”. Journal of Media Horizons, vol. 6, no. 3, Aug. 2025, pp. 2542-51,
>
> 3. The use of AI-generated comments was a deliberate methodological decision, not for convenience. While deceptive humor comments exist in the wild, systematically identifying which comments embed deceptive claims is extremely challenging, and reliable verification is often difficult and may involve safety risks. Controlled synthetic generation allowed us to systematically vary false claims, humor types, and languages in ways infeasible through natural collection, enabling controlled cross-lingual and stylistic analyses. To ensure human-like quality and representativeness, we employed a careful human-in-the-loop refinement and feedback process, producing comments that closely resemble real-world humorous misinformation. Supporting this, prior studies and reports (see Response 2) show that humorous framing is increasingly used in social media and online misinformation, including pandemic-era jokes that exaggerate everyday experiences while subtly linking to false claims. These real-world examples exhibit patterns similar to those in DHD, indicating that our dataset effectively captures realistic stylistic mechanisms of deceptive humor while maintaining experimental control.
>
> 4. We explicitly acknowledged this in the revised version's Limitations and Future Work sections that DHD is currently based solely on India-based fact-checking sources (AltNews, Boom FactCheck, FactChecker.in, and FACTLY). Consequently, the dataset may reflect cultural, linguistic, and political perspectives specific to India, which could influence the interpretation of humorous misinformation. Extending DHD to additional countries and cultural contexts is an important direction for future work, enabling evaluation of how deceptive humor operates across diverse linguistic and geopolitical settings.

---

> ### Author Response · Authors · 2025-11-16
> **Rebuttal by Authors**
>
> 5. We explored existing humor classification models; however, official weights for Naughtyformer and the XAI-based humor style classification work are not publicly available. To provide a comparison, we experimented with publicly available models under ideal conditions. While these models were trained on binary humor detection tasks and are not fully aligned with our multi-dimensional Satire Level and Humor Attribute annotations, the results are as follows:
>
>
> | Model | Task | Accuracy | M F1 | Wt F1 | Pear |
> |-------|------|----------|----------|-------------|---------|
> | CardiffNLP/twitter-roberta-base-irony (EMNLP 2020 Findings) | Satire Level | 29.89 | 23.47 | 21.56 | 15.39 |
> | CardiffNLP/twitter-roberta-base-irony (EMNLP 2020 Findings) | Humor Attribute | 29.00 | 19.62 | 18.42 | – |
> | Humor-Research/humor-detection-the-naughtyformer-693 | Satire Level | 26.11 | 23.39 | 19.14 | 11.90 |
> | Humor-Research/humor-detection-the-naughtyformer-693 | Humor Attribute | 28.78 | 18.94 | 17.86 | – |
>
> These preliminary experiments indicate that existing humor classification models perform poorly on DHD, likely due to the mismatch between binary humor detection training and our multi-dimensional tasks. We have not included these results in the revised paper since the official models are not publicly available. Nevertheless, the experiments suggest that DHD presents a challenging and unique testbed for studying the stylistic mechanisms of deceptive humor.
>
> 6. Regarding the concern about AI-generated data, we acknowledge that synthetic data may not fully capture real-world patterns (as noted in the Limitations). However, the use of AI-generated comments was a deliberate methodological choice: systematically identifying real-world deceptive humor is extremely challenging, and controlled synthetic generation allows systematic variation in false claims, humor types, and languages while maintaining experimental control. Human-in-the-loop refinement ensures high-quality, realistic comments (see Response 3 for details).
>
> Concerning baseline model sizes, we agree that DHD is generated using ChatGPT-4o, a large LLM, while our baselines involve smaller models. However, recent studies and empirical observations indicate that small- to medium-scale LLMs can sometimes outperform larger LLMs on specific downstream tasks. To further support this, we added evaluation results using Mistral-7B and LLaMA-8B, which demonstrate that performance differences on DHD are not solely determined by model size. These results suggest that the task’s inherent difficulty and the dataset’s multi-dimensional annotations play a major role in the observed performance gaps.
>
> ---
> **Response to Ethics Review Flags:**
> We thank the reviewer for highlighting these important ethical considerations. We have updated Appendix B (Ethical Considerations) to explicitly address all flagged areas, including (1) clarifying the legal basis for data collection and the fully synthetic generation process, along with measures ensuring privacy, security, and compliance; (2) describing safeguards and enforcement mechanisms to mitigate potential misuse of the dataset; (3) providing explicit content warnings regarding potential bias, offensive material, and responsible usage; and (4) detailing annotator protections, including informed consent, fair compensation, workload limits, optional academic benefits, and adherence to institutional ethical guidelines. We kindly invite the reviewers to refer to the revised section for full details.
>
> We have sincerely addressed all concerns and remain open to discussing further improvements. We respectfully request reconsideration of the rating in light of these substantive revisions.

---

### Official Review · Reviewer_kjsY · 2025-10-31

**Soundness:** 3
**Presentation:** 3
**Contribution:** 3
**Rating:** 6
**Confidence:** 4

**Summary:**

The paper introduces Deceptive Humor as a new research area combining misinformation and computational humor. The authors came up with the Deceptive Humor Dataset (DHD), which is a synthetically generated corpus of humorous comments in English, Telugu, Hindi, Kannada, Tamil, and their code-mixed variants. Each instance is associated with a false claim and then annotated with satire levels and humor types. The authors benchmark this dataset by proposing DH-MTL, which is multi-task learning framework that jointly learns satire intensity and humor type.

**Strengths:**

- I like the new problem direction of deceptive humor. With social media being used in all aspects of our daily lives, it does influence our thinking a lot and this study which deals with how affective traits like humor are being used to manipulate/persuade humans is interesting to me.
- Not limiting the study to just the English language is also a good decision. Especially, incorporating code-mixed input since that is how most natural conversations take place for multilingual folks.
- The way humans are involved in the loop is also interesting. As the authors did not just ask humans to evaluate responses, rather used the feedback to iterate the models. I think this is especially useful for multilingual settings to incorporate the sense of "cultural humor".

**Weaknesses:**

Although the paper is well written and is easy to follow, there are still some weaknesses that the paper can improve upon:
- The paper defines deception as repeating known false claims. It assumes that if a joke includes a debunked statement (like “China made COVID as a bioweapon”), then the joke is deceptive. But mentioning a false claim is not the same as supporting it, especially when the statements can be satirical. That is, the dataset currently treats every mention of a false claim as deceptive, even when the joke is actually mocking or criticizing that claim. This could wrongly label satire or political humor as harmful misinformation.
- Enforcing the above point, The paper says deceptive humor is harmful but never explains what deceptive means in this context. Is it about the writer trying to fool people or how readers understand it or just the false facts in it? The notion of deception is confusion right now.
- Although synthetic data is the easiest and fastest way to get data, but there is a big issue, especially when using languages other than English and dealing with extremely subjective problems like satire and humor. Table 7 shows that languages like Hindi and Tamil score lower on cultural nuance, which suggests these examples don't fully capture real humor.

**Questions:**

- How do you tell apart jokes that make fun of false claims (like mocking conspiracy theories) from jokes that actually spread them? Could DHD mark the first type as deceptive (which would be a mistake)? Same for satire, can the proposed method punish real satire that is actually criticizing the misinformation?

---

> ### Author Response · Authors · 2025-11-16
> **Rebuttal by Authors**
>
> Thank you for your thoughtful review and positive feedback. We sincerely appreciate your recognition of the novelty of deceptive humor as a research direction, our multilingual design, and the use of a human-in-the-loop refinement process. We have carefully considered all your comments, clarified the conceptual distinctions you highlighted, and incorporated the corresponding revisions into the updated version.
>
> ---
>
> 1. We agree that merely mentioning a false claim does not make a joke deceptive. In our work, deceptive humor refers only to humor that implicitly reinforces or normalizes a false claim, not humor that mocks or criticizes it. Annotators were trained with explicit guidelines to distinguish “mocking the claim” from “echoing the claim” as mentioned in the guidelines. During data creation, we filtered out jokes that ridiculed conspiracy theories or highlighted their absurdity; these were labeled as non-deceptive and excluded from the deceptive subset. Satirical or critical jokes were therefore not treated as deceptive, even if they referenced debunked statements. Only cases where humor frames a false narrative as plausible were included. Thus, DHD does not classify jokes mocking false claims as deceptive; instead, it focuses on instances where humor repeats or normalizes false claims. We have clarified this distinction in the revised introduction.
>
> 2. In our work, “deceptive humor” refers to humor that misleads or desensitizes audiences by making a false claim appear harmless, acceptable, or socially normal. Importantly, deception in our setting is effect-based rather than intent-based. We do not rely primarily on the author’s motivation, as humorous content is often copied, reshared, or remixed online, meaning the original intent is frequently unknown or irrelevant. Prior work also shows that misinformation can influence perceptions even when the creator did not intend to deceive.
>
> - For this reason, we majorly focus on the impact on readers: whether humorous framing lowers scrutiny and subtly reinforces the false narrative. The harm arises not from humor alone, but from how humor shapes audience perception of the embedded misinformation.
>
> 3. We agree that synthetic generation can struggle to capture cultural nuance, especially in subjective domains like humor. Our use of synthetic data was a deliberate methodological decision, not for convenience, but because real-world deceptive humor examples are extremely scarce, difficult to reliably verify, and often associated with safety risks. Controlled synthetic generation enabled us to systematically vary false claims, humor types, and languages in a way that would be infeasible through natural data collection.
>
> - To mitigate the limitations you highlighted, we incorporated a human-in-the-loop refinement process across all languages. Native speakers provided iterative feedback after each generation round, which significantly improved cultural specificity and contextual accuracy. The relatively lower nuance scores in Hindi and Tamil reflect the inherent difficulty of modeling humor in those languages rather than a lack of refinement. Importantly, the moderate-to-substantial IAA and expert validation indicate that the dataset maintains strong linguistic and conceptual reliability.
>
> - As also acknowledged in our Limitations section, synthetic data cannot fully replicate the richness of real-world humor; however, it offers a controlled and reproducible starting point for studying deceptive humor in multilingual settings where validated natural data is extremely difficult to collect, as it requires the complete contextual knowledge regarding the fabricated claim.
>
> 4. We distinguish between jokes that mock a false claim and those that reinforce it through humor. In DHD, a joke is considered deceptive only when the humorous framing implicitly echoes, normalizes, or lends plausibility to the false claim. If the joke contradicts, ridicules, or exposes the absurdity of the claim, it is annotated as non-deceptive.
>
> - Annotators were trained with explicit guidelines and examples illustrating this distinction (“mocking the claim” vs. “echoing the claim”). As a result, satire that criticizes or undermines misinformation is not labeled as deceptive, even when it references the same false narrative. This ensures that DHD does not penalize genuine satire or political humor.
>
> - For example, jokes like “China products don’t last long, except COVID” reinforce the bioweapon claim through humor, whereas jokes that highlight the absurdity of the claim would be marked non-deceptive. Our human-in-the-loop refinement process consistently filtered out such critical or mocking cases during data creation.
>
> Thus, DHD does not misclassify jokes mocking false claims as deceptive, and the proposed method does not penalize real satire that criticizes misinformation.
>
> ---
>
> We remain open to further discussion and clarification, we sincerely appreciate your time and constructive feedback.

---

### Author Response · Authors · 2025-11-20
**Official Comment: Methodology Clarification**

Based on the reviewer's concerns regarding domain-specific adaptability, particularly the points raised by Reviewer z6ht (Q5) and Reviewer 8CL2 (Q4), we recognize that deceptive humor operates within nuanced linguistic, cultural, and misinformation domains for which no openly available task-specific models currently exist. Since deceptive humor is an underexplored research space, existing pretrained or fine-tuned models do not adequately capture these domain-dependent patterns (though we have provided fine-tuned results with openly available models such as `CardiffNLP/twitter-roberta-base-irony` and `Humor-Research/humor-detection-the-naughtyformer-693`, these models are not specifically designed for deceptive humor. They were trained for particular tasks, e.g., binary satire/irony classification, whereas our work involves multi-level satire (3 classes) and multi-class humor attributes (5 classes), and thus do not generalize to the full spectrum of deceptive humor phenomena).

To directly address these concerns, we have updated DH-MTL into a two-stage training framework designed to enhance domain grounding and improve generalization across languages and humor styles.

### **Stage 1: Domain Adaptation**

All encoder layers and both task-specific heads are jointly optimized to specialize the pretrained transformer to the distributional, stylistic, and narrative properties of deceptive humor. This stage enables the model to internalize, including:

- The interaction between false claim distortion and humor mechanisms
- Cultural and multilingual stylistic cues
- Subtle differences across code-mixed inputs

### **Stage 2: Task-Specific Fine-Tuning**

After domain alignment stabilizes, we apply gradual unfreezing. Training begins with only the top encoder layers being trainable, and deeper layers are progressively unfrozen in successive phases. This curriculum shifts the model from broad domain alignment to precise task reasoning, improving sensitivity to fine-grained humor attributes and subtle satire intensity distinctions while preserving foundational linguistic knowledge.

We also include detailed ablation studies and hyperparameter sensitivity analyses, demonstrating the effectiveness and robustness of the two-stage framework. While this approach may not achieve absolute SOTA on every metric, it significantly improves performance on low-frequency and challenging classes, which prior baselines, including the previous version of DH-MTL and LLMs, struggled to model reliably.

We sincerely thank the reviewers for highlighting the importance of domain-specific considerations. We believe that the revised two-stage DH-MTL clarifies and strengthens the methodological rigor and directly addresses the concerns raised. We remain open to providing further details or incorporating additional refinements.

We have sincerely addressed all the concerns raised by the reviewers and respectfully request reconsideration of the ratings in light of these substantive revisions.

---

### Author Response · Authors · 2025-11-23
**Author's Official Response to Reviewer Comments (1/2)**

We sincerely thank all reviewers (kjsY, z6ht, 8CL2, rXsS) for their thoughtful feedback and for recognizing deceptive humor as an important and emerging research direction. In addition to point-by-point responses, we provide a unified clarification of central themes raised across reviews: dataset design, task relevance, and methodological rigor. The revised version incorporates substantial conceptual, methodological, and empirical improvements aligned with these concerns, which we invite the reviewers to examine in detail.

---

### **1. Why Synthetic + Human-in-the-Loop (HITL) Data Is Necessary**
### (Addresses: 8CL2, z6ht, rXsS)

Real-world deceptive humor cannot be reliably collected because intent is unobservable. A humorous false claim may be:

- mocking the claim (non-deceptive), or
- implicitly reinforcing it (deceptive).

Surface text alone cannot distinguish these cases, making real-world extraction fundamentally unreliable.

Our HITL pipeline fixes the false claim and systematically varies the humor style through LLM generation, followed by native-speaker refinement. This controlled setup allows us to isolate the linguistic mechanisms, such as exaggeration, irony, and framing, that modulate how misinformation is perceived, which is impossible with noisy web data.

To ensure cultural grounding, annotators across Telugu, Hindi, Kannada, Tamil, and English, including postgraduate students, PhD researchers, and a senior faculty member, iteratively filtered and refined generated batches for contextual plausibility and label correctness.

**Crucially, our evaluation is grounded in human judgment, not synthetic artifacts**. The **Test Set (N = 900) was manually annotated by humans, achieving moderate-to-substantial Inter-Annotator Agreement (Fleiss’ κ = 0.62)** (see Appendix K). This ensures that while our training data is synthetic-seeded, our benchmarking metrics reflect performance on a human-verified gold standard, not model-generated noise.

We acknowledge that synthetic data cannot fully replicate real-world humorous misinformation. However, given the identification challenges, safety constraints, and intent ambiguity inherent in wild data, a controlled HITL setting provides the most scientifically valid starting point for modeling deceptive humor.

---

### **2. Clear Definition of Deceptive Humor**
### (Addresses: kjsY)

We have updated the definition of deceptive humor in the revised version as humor that reinforces, normalizes, or lends plausibility to a false claim. Any content that mocks, criticizes, or exposes the false claim is explicitly excluded. Annotators were instructed and trained to identify such instances and mark them as non-deceptive, removing them during HITL refinement. This ensures that DHD focuses on humor as a vector for misinformation rather than satire used to debunk false claims.

---

### **3. Why Satire Level and Humor Attributes Matter**
### (Addresses: z6ht)

The reason is that in humorous contexts, misinformation does not operate through explicit factual claims. Instead, its effect arises from how humor reshapes the interpretation of the false claim. The scientific problem is therefore not detecting the claim itself, but modeling how humor modulates its impact.

**Why a traditional misinformation label is insufficient**

- In deceptive humor, the factual status of the claim (true/false) is not the key factor.
- Readers may already recognize the claim as false, yet humor can still increase acceptance, reduce skepticism, or normalize the narrative.
- A binary misinformation label would capture only the content, missing the mechanisms that make the claim persuasive when framed humorously.

**Why our tasks directly model misinformation mechanisms**

- Satire Level measures how strongly humor hides, softens, or exposes the false claim. This influences whether a reader interprets the claim as plausible, harmless, or obviously untrue.
- Humor Attributes capture narrative strategies, such as irony, absurdity, and wordplay, that shape cognitive and emotional processing of the claim.

These dimensions directly determine whether humorous misinformation passes unnoticed, is shared, or is misinterpreted.

Thus, we did not include traditional misinformation labels because deceptive humor operates not through factual ambiguity but through pragmatic framing, subtlety, and stylistic modulation. In our case, Satire Level and Humor Attributes are precisely the misinformation-relevant variables that allow modeling these mechanisms.

---

### **4. Expanded Baselines with Modern 7B and 8B Models**
### (Addresses: z6ht)

We added evaluations on state-of-the-art models, including **LLaMA-3.1-8B-Instruct** and **Mistral-7B-v0.3-Instruct**.

Results show that even these instruction-tuned 7B, 8B models struggle to reliably detect or characterize deceptive humor. This confirms that the difficulty is inherent to the **domain complexity**, rather than the scale or quality of baseline models.

---

> ### Author Response · Authors · 2025-11-23
> **Author's Official Response to Reviewer Comments (2/2)**
>
> ### **5. Methodological Rigor: Two-Stage DH-MTL Framework with Adaptive Loss Weighting**
> ### (Addresses: 8CL2, z6ht)
>
> We upgraded DH-MTL into a two-stage training framework:
>
> **Stage 1 (Domain Adaptation):** Aligns the encoder and task heads with stylistic, cultural, and code-mixed properties of deceptive humor.
>
> **Stage 2 (Gradual Unfreezing):** Specializes deeper layers for Satire Level (ordinal) and Humor Attributes (categorical) while preserving foundational linguistic knowledge.
>
> **Joint Multi-Task Optimization:**
> The model jointly optimizes Satire Level and Humor Attribute tasks with a weighted combination of their losses:
>
> ```math
> L_total = w_sat * L_sat + w_hum * L_hum
> ```
>
> Adaptive weighting allows the model to adaptively balance the contributions of the two tasks during training.
>
> **Ablation Insights:**
> - Fixing the Satire Level weight significantly reduces performance on Satire Level metrics, and moderately affects Humor Attributes.
> - Fixing the Humor Attribute weight reduces performance on Humor Attributes, while Satire Level is less affected.
> - Fixing both weights produces the largest drop across all metrics.
>
> These results highlight that **adaptive loss weighting is essential** for robust multi-task learning. It ensures the model generalizes across all classes, including underrepresented labels, and captures both fine-grained stylistic and intensity distinctions in deceptive humor, something baseline LLMs struggle with.
>
> ---
>
> ### **6. Real-World Grounding of Deceptive Humor**
> ### (Addresses: rXsS)
>
> To demonstrate the real-world relevance of deceptive humor, we added two key analyses:
>
> - **Appendix G: Behavioral Evidence** – Summarizes studies showing that humor can lower epistemic vigilance, reduce scrutiny, and increase belief in false claims. These findings directly support our motivation for modeling how humor affects misinformation perception.
>
> - **Appendix H: Case Studies** – Provides concrete examples from geopolitical conflicts, pandemic rumor cycles, and meme-driven misinformation, illustrating that deceptive humor is a documented mechanism for spreading false claims. These cases contextualize our tasks and show that Satire Level and Humor Attributes capture phenomena observed in real-world information ecosystems.
>
> ---
>
> We have carefully addressed all concerns raised by the reviewers and provided detailed clarifications, methodological improvements, and empirical validations. We remain fully open to further discussion and are happy to provide any additional insights or clarifications as needed.
>
> Sincerely,
>
> The Authors

---

### Author Response · Authors · 2025-12-01
**Author's Official Response: Summary of Revisions**

We thank all reviewers for their insightful comments, thoughtful feedback, and for recognizing deceptive humor as an important and emerging research direction. We have revised the paper to address the reviewer's concerns. Below, we summarize the major revisions:

---

1. **Clarified the Necessity of Synthetic + Human-in-the-Loop (HITL) Data:**
   Explained that real-world deceptive humor cannot be reliably collected due to unobservable intent. To address this, we adopted a controlled generation + HITL refinement pipeline to systematically study this emerging phenomenon. The HITL process ensures quality, cultural relevance, and contextual plausibility across languages, producing a multilingual, code-mixed dataset. The **Test Set (N = 900)** is fully human-annotated with moderate-to-substantial IAA (**Fleiss’ κ = 0.62**). The full HITL pipeline is detailed in Appendix I.

2. **Clarified the Definition of Deceptive Humor:**
   Defined deceptive humor as humor that reinforces or lends plausibility to a false claim rather than mocking or criticizing it, with updates to the annotation protocol and HITL guidelines (Section 3.1, Appendix C). We emphasize that this work introduces a novel and scientifically rigorous approach: the HITL synthetic data pipeline is a methodologically necessary innovation for safely studying deceptive humor that is otherwise unobservable, and the multilingual, code-mixed dataset ensures broad multilingual applicability and insights beyond English.

3. **Strengthened the Motivation for Satire Level and Humor Attributes:**
    Explained that humor modulates how false claims are perceived, which binary misinformation labels cannot capture. Satire Level and Humor Attributes measure the intensity and stylistic strategies (e.g., irony, absurdity, wordplay) that influence plausibility and audience interpretation (Section 3.4, Appendix D).

4. **Expanded Baselines with Larger Models:**
   Added evaluations on **Mistral-7B-v0.3-Instruct and LLaMA-3.1-8B-Instruct**, showing that deceptive humor remains challenging even for large models (Section 5, Table 3, Appendix F).

5. **Upgraded DH-MTL to a Two-Stage Training Framework:**
   Introduced a two-stage framework: Stage 1 domain adaptation, Stage 2 task-specific fine-tuning with gradual unfreezing and adaptive loss weighting; methodological details, ablations, and parameter sensitivity analyses added (Section 5, Appendices D-E).

6. **Real-World Grounding of Deceptive Humor:**
   Added behavioral evidence and real-world case studies demonstrating how humor facilitates the spread and acceptance of false claims (Appendices G-H).


---

**Response to Ethics Review Flags:**
We thank the reviewer who raised ethical concerns for highlighting these important points. We have expanded **Appendix B (Ethical Considerations)** to explicitly address all flagged areas, including:
- clarifying the legal basis and fully synthetic data generation process with privacy safeguards,
- mitigation strategies and enforcement mechanisms to prevent dataset misuse,
- explicit content warnings regarding potential bias and offensive material, and
- annotator protections, including informed consent, fair compensation, workload limits, optional academic benefit, and adherence to institutional ethical guidelines.

We once again sincerely thank the reviewers for their insightful comments. The revisions demonstrate the rigor and relevance of our work.

Sincerely,

The Authors

---

### Meta-Review · Area_Chair_efaZ · 2025-12-25

**Summary:**

The reviews were mixed initially, with scores of 6, 2, 4, 4. The reviewers generally agreed on the novelty of the research direction, the multilingual settings, and the human-in-the-loop refinement process. However, the primary concerns regarding the validity of synthetic data and the task formulation are not well addressed. There is a misalignment between the motivation (misinformation detection) and the proposed tasks (stylistic humor classification). Despite a comprehensive rebuttal, the absence of real-world validation prevents the paper from meeting the acceptance threshold. Moreover, the reviewers noted the data sources were exclusively India-based, which contradicts the broad multilingual claims. The authors acknowledged this valid limitation and listed it for "future work," but the bias in the current dataset remains unresolved.

**Reviewer Concerns:**

Concerns Effectively Addressed:

- The reviewer kjsY was concerned that the dataset might conflate "mocking a false claim" (satire) with "supporting a false claim" (deception). The authors effectively clarified that they filtered out mocking instances and provided explicit guidelines to annotators to only include humor that reinforces or normalizes the claim.
- The rebuttal provides explanations related to pragmatics, culture, and code-mixing.

Outstanding or Partially Unresolved Concerns:

- While the Human-in-the-Loop process was highlighted, the reviewers (kjsY, rXsS, z6ht) noted that synthetic data struggles with deep cultural nuance. The authors admitted this cannot fully replicate "wild" data, leaving the concern valid but mitigated.

- Both reviewers rXsS and 8CL2 thought that there's a disconnection between Motivation and Task. The reviewers felt that classifying "Satire Level" does not equal "Detecting Misinformation."

- The reviewer rXsS noted the data sources were exclusively India-based. The authors acknowledged this valid limitation and listed it for "future work," but the bias in the current dataset remains unresolved.

**Reviewer Scores:**

Reviewer kjsY: Likely would maintain their score. They were positive about the direction and satisfied with the definitions, but the methodological flaws raised by other reviewers would likely prevent them from giving a higher score.

Reviewer rXsS: Might stick to their rejection based on the geographical bias of the data sources and synthetic validity issue.

Reviewer z6ht: Likely would remain score 4 or slightly increase the score to 6. While the rebuttal addressed the "performance drop" logic, the reviewer's concern about the representativeness of synthetic code-mixed data is not well addressed.

Reviewer 8CL2: Would likely settle on a weak rejection.

---

### Decision · Program_Chairs · 2026-01-26

Reject